# WHICH EXAMPLES TO ANNOTATE FOR IN-CONTEXT LEARNING? TOWARDS EFFECTIVE AND EFFICIENT SELECTION

## ABSTRACT

Large Language Models (LLMs) can adapt to new tasks via in-context learning (ICL). ICL is efficient as it does not require any parameter updates to the trained LLM, but only few annotated examples as input for the LLM. In this work, we investigate an active learning approach for ICL, where there is a limited budget for annotating examples. We propose a model-adaptive optimization-free algorithm, termed ADAICL, which identifies examples that the model is uncertain about, and performs semantic diversity-based example selection. Diversity-based sampling improves overall effectiveness, while uncertainty sampling improves budget efficiency and helps the LLM learn new information. Moreover, ADAICL poses its sampling strategy as a Maximum Coverage problem, that dynamically adapts based on the model's feedback and can be approximately solved via greedy algorithms. Extensive experiments on nine datasets and seven LLMs show that ADAICL improves performance by 4.4% accuracy points over SOTA (7.7% relative improvement), is up to $3\times$ more budget-efficient than performing annotations uniformly at random, while it outperforms SOTA with $2\times$ fewer ICL examples.

## 1 INTRODUCTION

Large Language Models (LLMs) have shown remarkable performance in various natural language tasks. One of the LLMs' advantages is their ability to perform few-shot learning (Brown et al., 2020), where they can adapt to new tasks, e.g., topic classification or sentiment prediction, via in-context learning (ICL). ICL uses few-shot labeled examples $\{(x_i, y_i)\}_{i=1}^k$, e.g., $(x_i, y_i) = $ (Amazing movie!, positive), to construct a prompt $P$. Prompt $P$ is used as a new input to the LLM, e.g., "Amazing movie!: positive \n Awful acting: negative \n Terrible movie:", before making predictions for the query $x_{\text{test}}$. The new input enables the LLM to infer the missing label $y_{\text{test}}$ by conditioning the generation on the few-shot examples. As semantically similar demonstrations to the test query improve ICL performance (Liu et al., 2021), it is common practice that a $k$-NN retriever is used to determine the $k$-nearest examples for each test query.

ICL is efficient as it does not require any parameter updates or fine-tuning, wherein users can leverage ICL to generate task-adaptive responses from black-box LLMs. However, ICL is sensitive to the input prompt (Lu et al., 2022) (the art of constructing successful prompts is referred to as prompt engineering), where acquiring ground-truth labeling of the input demonstrations is important for good ICL performance (Yoo et al., 2022). Ground-truth labeling requires expert annotators and can be costly, especially for tasks in which the annotators need to provide elaborate responses (Wei et al., 2022b). Apart from lowering the labeling cost, carefully reducing the number of the ICL examples can benefit inference costs and the LLM's input context length requirements. Consequently, we study the following problem: *Given a budget B, which examples do we select to annotate and include in the prompt of ICL?*

Given an unlabeled set (pool) where we can draw ICL examples from, the above selection problem resembles a typical active learning setting Settles (2009); Zhang et al. (2022b). Active learning selects *informative* examples, e.g., via uncertainty sampling (Lewis & Gale, 1994), which are used to improve the model's performance. Although traditional active learning involves model parameter updates, uncertainty sampling has been explored in an optimization-free manner for ICL with black-

box LLMs (Diao et al., 2023; Pitis et al., 2023). Yet, recent studies show that uncertainty sampling yields inferior performance in comparison to other approaches (Margatina et al., 2023) and thus, current methods rely on semantic diversity to determine which examples are the most informative. For example, AutoCoT (Zhang et al., 2023b) performs clustering based on semantic similarity and selects the most representative examples of each cluster. In order to adapt the selection based on the LLM used, Vote$k$ (Su et al., 2022) selects diverse examples with respect to the LLM's feedback, i.e., examples that the model is both uncertain and confident about. However, these approaches do not consider which examples help the LLM learn new information (uncertainty sampling) and may waste resources for annotating examples whose answers are already within the model's knowledge.

To overcome the aforementioned limitations of active learning for ICL, we pair uncertainty-based sampling with diversity-based sampling. Combining uncertainty and diversity sampling has been previously used for finetuned-based NLP (Yu et al., 2022; 2023), but it is not purposed for non-parametric classifiers, similar to how ICL works (Han et al., 2023; Bai et al., 2023). To best combine the two sampling strategies, we propose a model-adaptive optimization-free method, termed ADAICL, which is tailored to retrieval-based $k$-shot ICL. ADAICL uses the LLM's feedback (output probabilities) to identify the examples that the model is most uncertain about (*hard examples*).

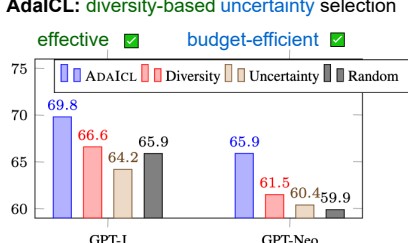

Figure 1: ADAICL effectively combines diversity and uncertainty sampling, outperforming other strategies in the low-resource scenario, averaged over seven datasets. Here, the budget is 20 annotations for retrieval-based 5-shot ICL.

The algorithm then identifies different semantic regions of hard examples, with the goal to select the most representative examples within each region. Diversity-based sampling is posed as the well-studied Maximum Coverage problem (MAXCOVER), which can be approximately solved via greedy algorithms. ADAICL performs subset selection (set of hard examples) and maximizes a submodular function (MAXCOVER) to capture the effect of different examples to the non-parametric ICL; a framework successfully employed for active learning with non-parametric $k$NN classifiers (Wei et al., 2015).

By selecting representative examples of diverse hard regions, ADAICL aims for *effectiveness*, so that it helps the LLM learn information that it does not already know. Moreover, ADAICL is *efficient* and results in budget savings by omitting the selection of examples that the model already knows how to tackle (easy examples). Finally, we show that ADAICL's uncertainty sampling improves the LLM's calibration (Jiang et al., 2021; Zhao et al., 2021), i.e., how model's confidence aligns with epistemic uncertainty, which measures how well the model understands the task.

We conduct experiments on nine datasets across five NLP tasks (topic classification, sentiment analysis, natural language inference, summarization, and math reasoning) with seven LLMs of varying sizes (1.3B to 65B parameters), including LLMs such as GPT-J (Wang & Komatsuzaki, 2021), Mosaic (MosaicML, 2023), Falcon (Penedo et al., 2023), and LLaMa (Touvron et al., 2023). Experimental results show that (i) ADAICL is effective with an average performance improvement of 4.4% accuracy points over SOTA (Figure 1), (ii) ADAICL is robust and achieves up to $3\times$ budget savings over random annotation, while it needs $2\times$ fewer ICL examples than SOTA (Section 6.2), and (iii) ADAICL improves the calibration of the LLM's predictions (Section 6.3).

## 2 RELATED WORK

**Active Learning for NLP**. Active learning (Settles, 2009) for NLP has been well-studied (Zhang et al., 2022b) with applications to text classification (Schröder & Niekler, 2020), machine translation (Haffari et al., 2009), and name entity recognition Erdmann et al. (2019), among others. Ein-Dor et al. (2020) studied the application of traditional active learning techniques (Lewis & Gale, 1994; Sener & Savarese, 2018) for BERT pretrained models (Devlin et al., 2019), with many works following up (Margatina et al., 2021; Schröder et al., 2022) and (Yu et al., 2022; 2023). These approaches fine-tune the model during different active learning rounds, which allows the model to incorporate information from the newly labeled examples into its parameters to gradually improve its predic-

tions. However, LLMs with billions of parameters are used for ICL. In this case, computing gradient updates is costly and requires additional fine-tuning for every new task. Furthermore, ICL acts as a nonparametric kernel regression (Han et al., 2023; Bai et al., 2023). Designing active learning for non-parametric classifiers has been recently highlighted to be challenging (Rittler & Chaudhuri, 2023), as the assumption that new information is incorporated into the model's parameters does not hold.

**Active Learning for ICL**. In this work, we focus on the "cold-start" problem, similar to (Su et al., 2022), where we are given an unlabeled set to select examples from. Most of the current approaches of active learning for ICL (Zhang et al., 2022a; Li & Qiu, 2023; Nguyen & Wong, 2023; Shum et al., 2023; Ma et al., 2023) assume a high-resource setting, where a large set of ICL examples is already annotated (validation set). The validation set is leveraged for measuring the informativeness of each individual example as well as for hyperparameter tuning. For example, Zhang et al. (2022a) employs reinforcement learning, which requires one set of labeled examples for policy training and another set of labeled examples for reward estimation. This limits the applicability in practical low-resource scenarios (Perez et al., 2021), where annotations are costly to obtain.

## 3 PROBLEM STATEMENT & MOTIVATION

Given an unlabeled set $\mathcal{U} = \{x_i\}_{i=1}^N$ and a fixed budget $B \in \mathbb{Z}^+$, the goal is to select a subset $\mathcal{L} \subset \mathcal{U}$, where $\mathcal{L} = \{(x_i, y_i)\}_{i=1}^B$ contains $B$ selected examples that are annotated. Due to token-length limits or inference cost considerations, we consider a $k$-shot ICL inference, where set $\mathcal{L}$ is used to draw $k$-shot ICL examples from ($k < B$). The $k$-shot examples are used to construct a new prompt $P$ as input to the LLM by

$$P = \pi(x_1, y_1) \oplus \pi(x_2, y_2) \oplus \cdots \oplus \pi(x_k, y_k) \oplus \pi(x_{\text{test}}, *). \tag{1}$$

Template $\pi$ denotes a natural language verbalization for each demonstration $(x, y)$ and it also expresses how the labels $y$ map to the target tokens. The selected $B$ examples for set $\mathcal{L}$ should maximize the ICL model performance on the testing set.

To determine which ICL examples to use (and in which order), a $k$-NN retriever selects the top-$k$ examples from $\mathcal{L}$, e.g., $(x_k, y_k)$, for a test instance $x_{\text{test}}$ based on the similarity between $x_i \in \mathcal{L}$ and $x_{\text{test}}$ over a semantic space $\mathcal{S}$ (the order is determined by similarity scores). Employing a $k$-NN retriever for ICL example selection has demonstrated superior performance over random or fixed example selection (Su et al., 2022; Margatina et al., 2023; Liu et al., 2021). During inference, we use third-party embedding models, such as SBERT (Reimers & Gurevych, 2019), –instead of the LLM $M$ itself– for retrieving $k$-shot ICL examples over $\mathcal{S}$ due to their applicability in practice. First, small-scale embedding models, such as SBERT, are much faster during computations and inference, compared to the LLMs with billions of parameters. Second, for many black-box LLMs, we do not have access to their intermediate layers (or parameters) but only to their outputs (predictions and token log probabilities). We illustrate the overall problem setting in Figure 2.

**Motivation**. To understand the impact of the ICL examples on model predictions, we express ICL inference as a non-parametric kernel regression, following the theoretical works from Han et al. (2023); Bai et al. (2023). The prediction for the test instance $x_{\text{test}}$ is related to

$$\tilde{y}_{\text{test}} = \frac{\sum_{i=1}^k y_i K_{\mathcal{D}}(x_{\text{test}}, x_i)}{\sum_{i=1}^k K_{\mathcal{D}}(x_{\text{test}}, x_i)}, \tag{2}$$

where $K_{\mathcal{D}}(x_{\text{test}}, x_i)$ is a kernel that measures the similarity between $x_{\text{test}}$ with each of the $k$-shot retrieved instance $x_i$, which depends on the pretraining data distribution $\mathcal{D}$ for model $M$. The importance of the $k$-NN retriever in the lens of Equation 2 becomes clear: it fetches *semantically similar* examples to compute $\tilde{y}_{\text{test}}$ by regressing over their labels $y_i$. However, it does not account for which examples help the model learn the task to a larger extent, which depends on $K_{\mathcal{D}}$ and is infeasible to determine for pre-trained LLMs as there is usually no direct access to $\mathcal{D}$.

ICL acts similar to non-parametric $k$NN classifiers (Equation 2) and designing active learning strategies for such classifiers has been recently highlighted to be challenging (Rittler & Chaudhuri, 2023). New information cannot be directly incorporated into the model's parameters, as it is typically assumed in other works for finetuned-based NLP (Margatina et al., 2021).

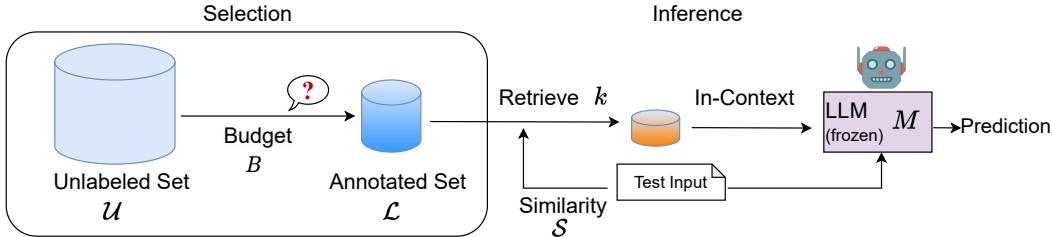

Figure 2: Our studied problem setting. Given an unlabeled set $\mathcal{U}$ and a fixed budget $B$, the goal is to select the $B$ most informative examples for annotation (set $\mathcal{L}$), which are used to maximize ICL performance with an LLM $M$. During ICL inference, a $k$-NN retriever based on a similarity space $\mathcal{S}$ determines the $k$-shot demonstrations for each test instance.

## 4  ADAPTIVE EXAMPLE ANNOTATION FOR ICL

*Can we improve example selection for ICL despite the aforementioned challenges?* We answer affirmatively by putting forth a **diversity-based uncertainty sampling** strategy, termed ADAICL, which adaptively identifies semantically different examples that help the model learn new information about the task. With diversity sampling, we seek **effectiveness**, i.e., good generalization during inference. With uncertainty sampling, we seek **budget-efficiency**, selecting examples that the model is *truly uncertain* about. Although combining diversity and uncertainty sampling has been previously explored for finetuned NLP (Yu et al., 2022; 2023), ADAICL focuses on combining the two strategies in the best way possible for ICL. ADAICL follows the successful framework of prior work (Wei et al., 2015) for active learning with non-parametric classifiers (subset selection & submodular function maximization) to capture the effect of different examples during retrieval-based ICL inference, but can generalize to both classification and generation tasks.

The ADAICL algorithm works as follows (see Figure 3 for an overview). First, it uses the LLM's feedback to determine which examples the model is most uncertain about, which we refer to as *hard examples*. Then, it builds different regions of hard examples and the goal is to select the most representative example for each region. The regions are determined based on the semantic similarity space $\mathcal{S}$, apart from the LLM's feedback, and denser regions are considered as more important.

### 4.1  ADAICL-BASE: A $k$MEANS APPROACH

A straightforward solution to combine uncertainty and diversity sampliing is to perform $k$means clustering (MacQueen et al., 1967) over the identified hard examples for the model, where different clusters represent regions of hard examples.

To determine the set $\mathcal{U}_h$ of hard examples, we use the LLM's feedback as follows. We assume we are given a small initially annotated pool $\mathcal{L}_0$ for $k$-shot ICL (if $\mathcal{L}_0 = \emptyset$, we perform zero-shot ICL) and use the LLM $M$ to generate a prediction for each $x_i \in \mathcal{U}$. For classification problems, we compute the conditional probability of each class $y \in \mathcal{Y}$, and the label $\tilde{y}_i$ with the maximum conditional probability is returned as the prediction for $x_i$. Similar to prior works (Min et al., 2022a), we used the *negative loglikehood* of the predicted label as the model's uncertainty score $u_i$, where lower score means that the model is more certain for its prediction $\tilde{y}_i$. For generation problems, we average the scores of the generated tokens as the model's uncertainty score $u_i$. We sort the examples $x_i \in \mathcal{U}$ based on their uncertainty scores $u_i$, and select the top-$N_\theta$ out of $N$ total examples, which are collected to $\mathcal{U}_h$. Here, $N_\theta = \lfloor \theta N \rfloor$ and $\theta \in [0, 1]$ is a hyperparameter with default value $\theta = 0.5$, which is the portion of the examples that we consider as hard ones.

Then, we can select representative examples for each region by sampling the example closest to each of the cluster centroids. Here, the number of clusters for $k$means is $B$, so we sample as many examples as the budget $B$ allows. We refer to that approach as ADAICL-base and its algorithm is summarized in Appendix B.1. Yet, ADAICL-base suffers from the known limitations of $k$means clustering. It is sensitive to outlier examples, which may be selected to be annotated, under the assumption that the $B$ regions formed by $k$means are equally important, and it does not account for the effect of overlapping clusters. We provide a failing case that the selected examples may not help the model understand the task in Appendix B.1.

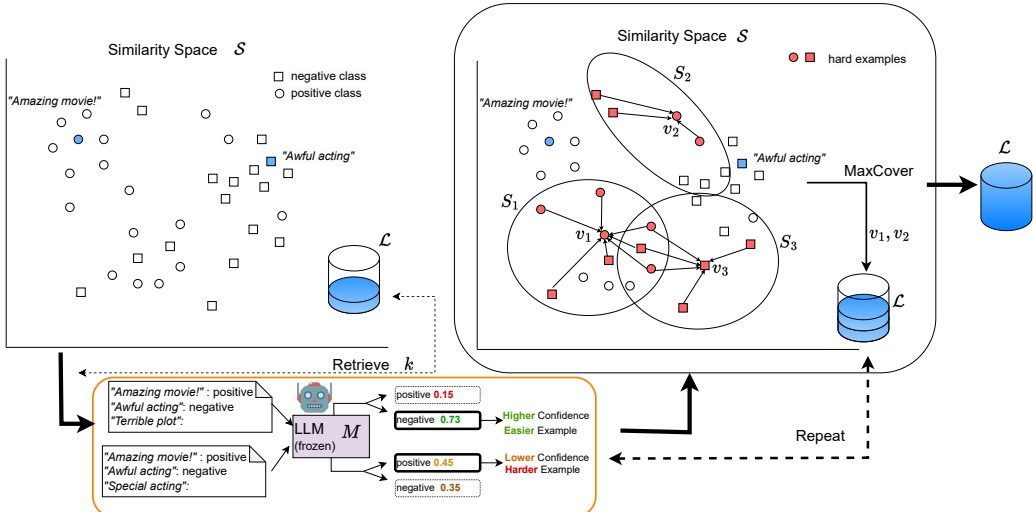

Figure 3: ADAICL algorithm. ADAICL uses $k$-shot ICL to determine which examples the model $M$ is uncertain for (hard examples). Then, it performs diversity-based uncertainty sampling over $\mathcal{S}$ by optimizing the MAXCOVER problem in Equation 3 via Algorithm 1 to identify the examples that help the model learn new information. The process is repeated until the budget $B$ is exhausted, and when done, it returns the annotated set $\mathcal{L}$.

## 4.2 ADAICL: SELECTION BY MAXIMUM COVERAGE

ADAICL overcomes the aforementioned limitations of ADAICL-base by quantifying whether each each example can convey new information to the model. ADAICL constructs semantic regions $S_i$ *around every* hard example $x_i$ and solves a Maximum Coverage (MAXCOVER) problem that accounts for information overlaps between different regions. MAXCOVER aims to select regions that cover the hardest examples, giving importance to denser regions and disregarding regions already covered (see Figure 3 for an overview). In addition, MAXCOVER does not require a finite set of label classes and can generalize to generation tasks – a limitation of (Wei et al., 2015) that studies active learning for non-parametric classifiers.

Formally, MAXCOVER takes $m$ sets $\{S_1, \ldots, S_m\}$ (regions that contain semantically similar examples) and a number $B$ as input. Each set includes some examples, e.g., $S_i = \{x_1, x_2, \ldots, x_n\}$ and different sets may have common examples, while the goal is to select the $B$ most representative sets that include (cover) as many examples as possible. We assume that if an example is marked as covered by another selected set, it conveys little new information to the model. In our setting, we are interested in hard examples, which we collect in the set $\mathcal{U}_h$ as previously explained. First, we discuss how we construct the regions and then we provide the MAXCOVER problem.

**Set Construction.** We represent the region $S_i$ around each example $x_i$ as its egonet. Initially, we build a global graph $\mathcal{G}_m$ as the $m$-nearest neighbors graph. The nearest neighbors are determined based on a semantic space $\mathcal{S}$ given by off-the-self encoders, such as SBERT (Reimers & Gurevych, 2019). We compute SBERT embeddings of each query $x_i$ and determine its neighbors based on cosine similarity of the embeddings. Graph $\mathcal{G}_m$ depends on the similarity space, while deriving sets $S_i$ from the global graph $\mathcal{G}_m$ depends on the LLM's feedback as we are interested in hard examples $x_i \in \mathcal{U}_h$ for the model.

For every hard node $v \in \mathcal{U}_h$, we construct its 1-hop egonet. We consider edges that *direct towards* $v$ from other hard nodes $v' \in \mathcal{U}_h$. This ensures that representative examples $x_i$, that are likely to be retrieved during ICL inference, have denser egonets. We experiment with both 1-hop and 2-hop set constructions. In the latter case, each node $v$ is represented by its egonet along with union of the egonets of its neighbors. One hyperparameter that controls the quality of the generated egonets is $m$, which is used during the construction of graph $\mathcal{G}_m$. In order to determine $m$, we employ a heuristic rule based on the desired maximum iterations $\hat{T}$ until the budget $B$ is exhausted, as well as the minimum number of hard examples $N_{\hat{\theta}}$ to be covered at each iteration. Due to space limitations, additional analysis of our approach are provided in Appendices B and F.

**Greedy Optimization.** The MAXCOVER problem is expressed as

$$\text{maximize} \sum_{x_i \in \mathcal{U}_h} c_i, \tag{3}$$

$$\text{where } c_j \in \{0,1\}, s_i \in \{0,1\}, \text{ and } \sum s_i \leq B, \sum_{x_j \in S_i} s_i \geq c_j. \tag{4}$$

Equation 3 maximizes the coverage of the hard examples $\mathcal{U}_h$, indicator variable $c_j$ denotes if example $x_j$ is covered, and $s_i$ denotes if set $S_i$ is selected. The goal is to select the examples that convey new information to the model (measured by the indicator $c_j$). Equation 4 ensures that we select at most $B$ sets and covered examples belong to at least one selected set (the hard examples covered in more sets are selected before others). To adjust the problem in our scenario, selecting set $S_i$, i.e., MAXCOVER marks $s_i = 1$, means that we select example $x_i$ for the annotated set $\mathcal{L}$.

The MAXCOVER problem is known to be NP-hard (Vazirani, 2001). A natural greedy solution for the MAXCOVER chooses sets according to one rule: at each stage, choose a set that contains the largest number of uncovered elements. This approximation algorithm is summarized below in Algorithm 1, and is well-known to approximately solve MAXCOVER and can be further improved due to its submodularity (Krause & Guestrin, 2005).

Note that the greedy algorithm is terminated when every hard example is covered, regardless of whether the budget $B$ is exhausted. In this case, the selected examples are added to the annotation set $\mathcal{L}$, and the model's feedback is reevaluated to define the new hard set

---

**Algorithm 1** Greedy approximation for MAXCOVER.

1: **Input:** Examples $\mathcal{U}_h$, Sets $\{S_1, \ldots, S_m\}$, Budget $B$.
2: **while** $B$ not exhausted **do**
3:     Pick the set that covers the most uncovered examples.
4:     Mark examples of the chosen set as covered.
5: **end while**

---

$\mathcal{U}'_h$. The iterative process is terminated when the total budget $B$ is exhausted. The overall framework is summarized in Figure 3, and the algorithm is summarized in Appendix B.2.

### 4.3 ADAICL+: DYNAMICALLY RE-WEIGHTED MAXCOVER

The greedy Algorithm 1 for ADAICL may cover all hard examples if the budget allows. However, this might include selecting sets that contain very few hard examples, e.g., outliers, or sets that belong to isolated sparse regions. ADAICL+ tackles this pitfall by a re-weighting schema for the MAXCOVER problem. Whenever a hard example is covered, instead of being marked as covered, ADAICL+ reduces its weight. By having new weights, dense regions with hard examples are preferred over sparse regions if their total weight is greater. We provide such a case in Appendix B.2.

Unfortunately, *dynamically* updating the weights of each example does not satisfy the submodularity property of MAXCOVER, which is satisfied for *fixed* weights. Nevertheless, such that we can use the greedy algorithm to approximate the optimal solution, we propose a re-weighting trick by reusing $\mathcal{U}_h$ multiple times. Specifically, we copy the set $\mathcal{U}_h$ multiple times, i.e., to $\mathcal{U}_h^0, \mathcal{U}_h^1$, etc., where different sets have different weights for their elements. If hard example $x_i$ is covered in $\mathcal{U}_h^0$, then we use its weights from the other sets. Formally, we optimize

$$\arg\max \sum_{t=0}^{\lfloor B/T \rfloor} \sum_{x_i^t \in U_h^t} w^t c_i^t, \tag{5}$$

We set the weights $w^t$, so that $w^t \approx w^t + w^{t+1} + w^{t+2} + \cdots$, which can be achieved by exponentially reducing the weights. In our case, we set $w^t = 10^{-t}$. In the beginning, every hard example of $\mathcal{U}_h^0$ has weight $w^0 = 1$. If one example is covered in $\mathcal{U}_h^0$, i.e., $c_i^0 = 1$, then its new weight is obtained from $\mathcal{U}_h^1$, where $w^1 = 0.1$. We introduce a new hyperparameter $T$, which denotes the desired total number of iterations until we exhaust the budget. At each iteration, we can annotate $\lfloor B/T \rfloor$ new examples by solving Equation 5 and then, the model $M$ re-evaluates its predictions. ADAICL+ algorithm is summarized in Appendix B.2.

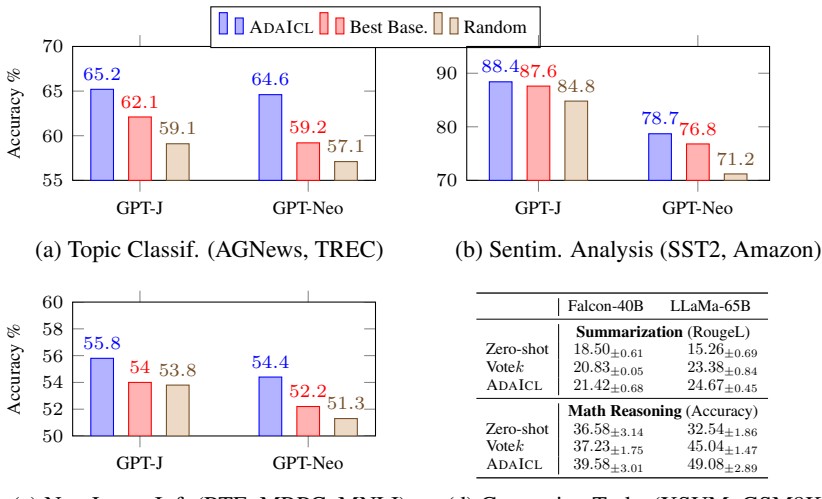

Figure 4: Performance comparison across different tasks with GPT-J (6B) and GPT-Neo (1.3B). "*Best Base.*" denotes the best baseline for the task. ADAICL performs the best, while for the classification tasks ADAICL-base is the second-best (full results in Appendix E.1).

## 5 EXPERIMENTAL SETTING

With our experimental analysis, we address the following research questions (RQs): *RQ1*. How does ADAICL compare with SOTA active learning strategies for ICL? *RQ2*. How efficient is ADAICL regarding labeling and inference costs? *RQ3*. How robust is ADAICL under different setups of the problem (Figure 2)? *RQ4*. Does ADAICL help the LLM understand the task?

**Datasets.** We performed empirical evaluation with nine NLP datasets that cover well-studied tasks, such as topic classification (AGNews (Zhang et al., 2015), TREC (Hovy et al., 2001)), sentiment analysis (SST2 (Socher et al., 2013), Amazon (McAuley & Leskovec, 2013)), natural language inference (RTE (Bentivogli et al., 2009), MRPC (Dolan et al., 2004), MNLI (Williams et al., 2018)), text summarization (XSUM (Narayan et al.)) and math reasoning (GSM8K (Cobbe et al., 2021)). We provide examples of these datasets and additional details in Appendix D.

**Baselines.** We use the following approaches as baselines for comparison: (i) **Random** performs random example selection for annotation. (ii) **Pseudo-labeling** uses the LLM to generate pseudo-labels for the unlabeled instances as additional annotated data. (iii) **Fast-vote**$k$ (Su et al., 2022) is a diversity-based sampling strategy that selects representative examples in the similarity space. (iv) **Vote**$k$ (Su et al., 2022) additionally accounts for the model's feedback. It sorts the examples based on the model's confidence scores and stratifies them into $B$ equally-sized buckets. It selects the top-scoring fast-vote$k$ example from each bucket. (v) **Hardest** resembles the uncertainty sampling strategy of active learning. The examples that the model is the most uncertain for are selected. (vi) Patron (Yu et al., 2023) that combines uncertainty and diversity sampling, but is designed for finetuned-based NLP. Additionally, we include (vii) **ADAICL-base** method (Section 4.1) as further ablations. All compared methods differ only on the "Selection" (Figure 2, while "Inference" is the same for all ($k$-shot retrieval-based ICL).

**Design Space.** As summarized in Figure 2, the design space includes the unlabeled set $\mathcal{U}$, the number of ICL examples $k$, the similarity space $\mathcal{S}$, the budget $B$, and the LLM $M$. We experiment with seven LLMs of varying sizes (1.3B to 65B parameters), including GPT, Mosaic, Falcon, and LLaMa model families, all of which are open-source and allow the reproducibility of our research. Unless otherwise stated, we set $k = 5$, $B = 20$ and we obtain embeddings in the similarity space via SBERT (Reimers & Gurevych, 2019). We experiment with inductive settings, where test instances come from an *unseen* set $\mathcal{U}_{\text{test}}$, but also for transductive settings, where test instances come from $\mathcal{U}$.

**ADAICL.** For the default problem setup, we construct 2-hop sets with $m = 5$ for ADAICL, and 1-hop sets with $m = 15$ for ADAICL+ via Equation 6. The default number of iterations $T$ for ADAICL+ is $T = 2$, while in the additive budget scenario we have $T = 1$. As the threshold hyper-

Table 2: Performance comparison across different retrieval and semantic similarity configurations.

| Retriever, $\mathcal{S} \rightarrow$ | SBERT-all-mpnet-base-v2 | | | RoBERTa-nli-large-mean-tokens | | | BERT-nli-large-cls-pool | | | Avg. |
|---|---|---|---|---|---|---|---|---|---|---|
| | TREC | SST2 | Amazon | TREC | SST2 | Amazon | TREC | SST2 | Amazon | |
| Pseudo-labeling | $48.56_{+6.33}$ | $69.13_{+3.87}$ | $70.96_{+3.35}$ | $33.98_{+3.68}$ | $74.08_{+4.40}$ | $81.11_{+4.14}$ | $41.27_{+4.24}$ | $77.47_{+1.60}$ | $81.63_{+2.49}$ | 64.24 |
| Random | $54.68_{+1.68}$ | $68.48_{+1.87}$ | $73.95_{+2.03}$ | $37.23_{+2.30}$ | $74.21_{+3.50}$ | $84.46_{+3.21}$ | $34.75_{+2.41}$ | $72.65_{+5.82}$ | $80.20_{+3.34}$ | 64.51 |
| Vote$k$ | $54.81_{+0.49}$ | $73.69_{+9.05}$ | $75.13_{+0.98}$ | $37.77_{+4.65}$ | $76.16_{+2.23}$ | $84.11_{+1.28}$ | $42.43_{+3.34}$ | $80.85_{+2.09}$ | $83.59_{+1.77}$ | 67.61 |
| ADAICL-base | $48.24_{+0.98}$ | $77.86_{+1.02}$ | $75.77_{+3.63}$ | $38.12_{+5.74}$ | $78.12_{+5.30}$ | $\underline{85.93}_{+2.30}$ | $38.15_{+3.10}$ | $78.64_{+2.78}$ | $\underline{85.80}_{+1.75}$ | 67.40 |
| **ADAICL** | $\underline{55.33}_{+2.57}$ | $\underline{79.68}_{+2.47}$ | $\underline{77.73}_{+2.23}$ | $\underline{39.06}_{+3.37}$ | $\underline{81.11}_{+1.50}$ | $85.15_{+0.55}$ | $\underline{44.06}_{+2.49}$ | $80.85_{+2.83}$ | $84.65_{+3.52}$ | **69.74** |

parameter $\theta$, we have $\theta = 0.5$, i.e., 50% of the examples are considered as hard. Hyper-parameter sensitivity studies that show ADAICL's robustness are presented in Appendix F.

## 6 RESULTS & ANALYSIS

### 6.1 RQ1: ADAICL IS EFFECTIVE

Figures 4a, 4b, and 4c show performance results for classification tasks with two different models GPT-J (6B) and GPT-Neo (1.3B). ADAICL is the method that achieves the best performance, with an improvement of up to 7.5% accuracy points over random selection. The overall improvement over the best baseline is 1.9% points for GPT-J and 3.2% for GPT-Neo, which shows that ADAICL is important for smaller sized LMs. The second best performing method for topic classification and sentiment analysis is ADAICL-base, which shows the importance of diversity-based uncertainty sampling for ICL. Figure 4d provides results for generation tasks. On the challenging reasoning tasks, ADAICL outperforms Vote$k$ and zero-shot ICL by 4.04% and 16.54% in accuracy, respectively.

### 6.2 RQ2 & RQ3: ADAICL IS EFFICIENT AND ROBUST

**Budget-Efficiency**. We experiment with a scenario similar to mini-batch active learning, where the budget increases in different steps and the retriever uses as many ICL annotated examples as the context-length limit allows. Figure 5 shows results when incrementing the budget with 10 more annotations (for 4 steps). ADAICL performs the best in all cases, where the average accuracy improvement over the best baseline that uses the model's feedback is 7.09% for topic classification, 1.86% for sentiment analysis, and 2.36% for natural language inference. Figure 5 also shows that for topic classification and sentiment analysis ADAICL

Figure 5: Multi-step results with GPT-Neo. Sweet point: the point at which we exceed the best performance achieved by random selection. Full results are in Appendix E.2.

exceeds the best performance achieved by random annotation with $3\times$ less budget.

**Impact of ICL examples**. Table 1 investigates ADAICL's efficiency with respect to the number of ICL examples used during inference (and annotated). As shown, ADAICL outperforms vote$k$ although it uses and annotates $2\times$ fewer ICL examples. This indicates that ADAICL identifies examples that help the LLM learn the task, while it can reduce the inference costs due to shorter input prompts. We provide a visualization of ADAICL's process in Appendix G.

Table 1: Impact of the number of ICL examples.

| | GPT-J (6B) | | Mosaic (7B) | |
|---|---|---|---|---|
| | AGNews | SST2 | AGNews | SST2 |
| Vote$k$ 5-shot | $53.61_{+7.72}$ | $72.89_{+11.67}$ | $71.04_{+8.12}$ | $80.98_{+5.26}$ |
| Vote$k$ 10-shot | $58.32_{+2.74}$ | $80.08_{+7.43}$ | $76.09_{+3.37}$ | $90.23_{+0.31}$ |
| ADAICL 5-shot | $67.44_{+4.57}$ | $83.98_{+1.10}$ | $77.20_{+3.42}$ | $89.58_{+1.75}$ |

$B = 5$ and $B = 10$ for 5-shot and 10-shot ICL, respectively.

**Retriever Effect**. Table 2 shows results when we use different off-the-shelf encoders for the similarity space $\mathcal{S}$, such as BERT (Devlin et al., 2019) and RoBERTa (Liu et al., 2019). As discussed in Section 3, it is difficult to approximate the true similarity between examples based on the pretraining distribution $\mathcal{D}$, and thus different encoders lead to different results. For example, SBERT achieves a maximum average performance of 55.33% and 77.73% for TREC and Amazon, respectively, while BERT achieves 44.06% and 85.80%. Despite the encoder choice, ADAICL performs overall the best as its diversity-based uncertainty sampling mitigates this effect.

**LLM Effect**. Figure 6 shows results when using different LLMs of similar sizes (6-7B parameters). The best performance is achieved for the Mosaic and GPT-J models. LLaMa does not display effective ICL capabilities and Falcon does not substantially improve with more annotated

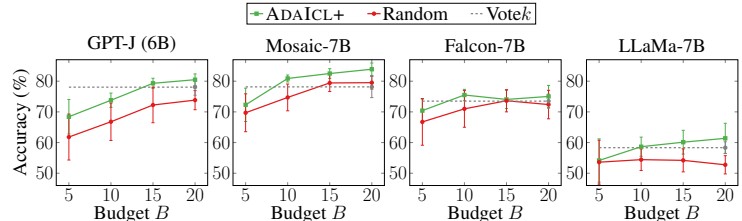

Figure 6: Average results over AGNews, TREC, SST2, and Amazon datasets for four LLMs with similar size.

examples. For Mosaic and GPT-J models, ADAICL outperforms Vote$k$ by 4.09% accuracy points, while the average accuracy improvement over random annotation for different budgets is 5.45% points.

### 6.3 RQ4: ADAICL IMPROVES CALIBRATION

One motivation behind uncertainty sampling is that it helps the LLM understand the task. Our hypothesis is that selecting examples that the LLM is over-confident for, i.e., easy examples, might not convey new information for the LLM to truly understand the task. We test our hypothesis by introducing a new variant of our method, termed ADAICL-easy, which constructs regions of hard examples *around easy examples* for MAXCOVER. To compare ADAICL-easy with our original ADAICL-hard, we compute the expected calibration error (ECE) (Guo et al., 2017) which quantifies the discrepancy between a model's predicted probabilities (how well it believes it understands the task) and observed outcomes (how well it actually solves the task). In addition, we visualize the discrepancy across probability bins by reliability plots, where any deviation from the straight diagonal indicates miscalibration and misunderstanding of the task.

As shown in Figure 7, "phase changes" in ECE could be observed for ADAICL-hard at different iterations and thus, improved calibration. We conjecture that the over-confident examples selected using ADAICL-easy lead to the bias towards making over-confident ICL predictions that are not always true. This is verified by the reliability plots for two snapshots: at iteration=7 for AGNews and at iteration=2 for SST2, where ADAICL-hard reduces the ECE from 0.20-0.30 to approximately 0.1. ADAICL-easy tends to make over-confident predictions (heavily skewed towards the right of the x-axis and large deviation from the diagonal), whereas ADAICL-hard produces more calibrated predictions, with more uniform probability distributions. We provide additional calibration analysis in Appendix E.5.

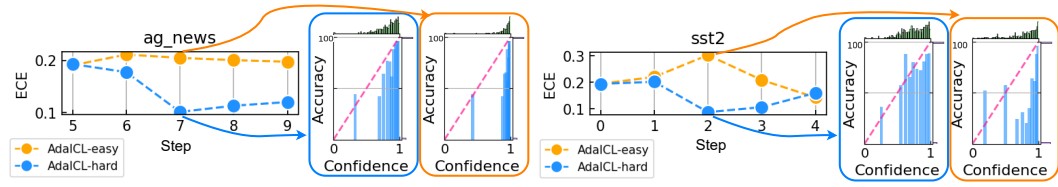

Figure 7: Expected calibration errors (ECE – the lower the better) and reliability plots for ADAICL when selecting easy (ADAICL-easy) and selecting hard (ADAICL-hard) examples.

## 7 CONCLUSIONS

In this work, we investigated budgeted example selection for annotation for ICL, a previously under-explored area. Seeking for an effective and efficient selection, we introduce ADAICL, a diversity-based uncertainty selection strategy. Diversity-based sampling improves effectiveness, while uncertainty sampling improves budget efficiency and helps the LLM learn new information about the task. Extensive experiments in low-resource settings show that ADAICL outperforms other approaches in five NLP tasks using seven different LLMs. Moreover, ADAICL can result into considerable budget savings, while it also needs fewer ICL examples during inference to achieve a given level of performance, reducing inference costs. Finally, our calibration analysis showed that ADAICL selects examples that lead to well-calibrated predictions.

# 8 REPRODUCIBILITY STATEMENT

The code of our ADAICL algorithm is is submitted as supplementary material. ADAICL's algorithmic steps are extensively summarized in Appendix B. For the datasets used, a complete description of the data processing steps is given in Appendices D.1 and D.2. Details of the experimental configurations are given in Appendix D.3.

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

APPENDIX

## A  EXTENDED RELATED WORKS

**ICL Mechanism**. In-context learning (ICL), also referred to as few-shot learning (Brown et al., 2020), has been shown to elicit reasoning capabilities of LLMs without any fine-tuning (Wei et al., 2022a; Bommasani et al., 2021; Dong et al., 2022). While ICL performance correlates to the pre-training data distribution (Xie et al., 2021; Hahn & Goyal, 2023; Shin et al., 2022; Razeghi et al., 2022; Chan et al., 2022) and improves with larger LMs (Bansal et al., 2023; Wei et al., 2023), recent works study how the transformer architecture (Vaswani et al., 2017) enables in-context learning (Akyürek et al., 2023; Olsson et al., 2022). Theoretical analyses and empirical studies show that ICL is a learning algorithm that acts as a linear regression (Akyürek et al., 2023; Garg et al., 2022; Von Oswald et al., 2023; Zhang et al., 2023a) and ridge (kernel) regression (Han et al., 2023; Bai et al., 2023) classifier. In this work, we leverage the recent connections of ICL with kernel regression to highlight the challenges of active learning for ICL.

**Prompt Influence for ICL**. Although ICL is widely used with LLMs, its success is not guaranteed as it is sensitive to the input prompts (Lu et al., 2022; Chen et al., 2022a). ICL tuning improves stability Min et al. (2022a); Chen et al. (2022b); Xu et al. (2023) but requires additional training data, while other works analyze how the prompt design and the semantics of the labels Min et al. (2022b); Yoo et al. (2022); Wang et al. (2022); Wei et al. (2023) affect ICL performance. More related to our setting, Liu et al. (2021),Rubin et al. (2022), Margatina et al. (2023), and Su et al. (2022) illustrate the importance of the $k$-NN retriever for ICL. Our work also highlights the importance of the $k$-NN retriever and provides new insights for improving ICL performance.

## B  ADAICL DETAILS

### B.1  ADAICL-BASE

Algorithm 2 presents the overall ADAICL-base algorithm. First, ADAICL-base uses the LLM's feedback to identify hard examples for the model based on their uncertainty scores. Then, it performs $k$means clustering where different clusters represent regions of hard examples and selects representative examples for each region by sampling the example closest to each of the cluster centroids. Here, the number of clusters for $k$means is $B$, so that we sample as many examples as the budget $B$ allows.

ADAICL-base suffers from the known limitations of $k$means clustering. It is sensitive to outlier examples, which may be selected to be annotated, assumes that the $B$ formed regions are equally important, and does not account for the effect of overlapping clusters. We provide such a failing case in Figure 8. Here, the annotated examples (with $B = 2$) do not effectively represent the semantic space and may not help the model understand the task.

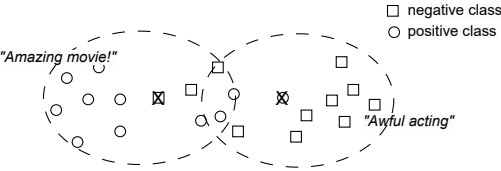

Figure 8: Failing case for $k$means clustering.

---

**Algorithm 2** ADAICL-base Algorithm.

---

1: **Input:** Model $M$, Unlabeled Set $\mathcal{U}$, Budget $B$, Similarity Space $\mathcal{S}$ for $k$-NN Retriever.
2: **Optional:** Initial set $\mathcal{L}_0$, else $\mathcal{L}_0 = \emptyset$.
3: **Hyperparameters:** threshold $\theta$.
4: **Output:** Annotated Set $\mathcal{L}$.

5: **for** $x_i \in \mathcal{U}$ **do**
6:     Retrieve (at most) $k$ examples from $\mathcal{L}$ based on similarity $\mathcal{S}$.
7:     Use model $M$ to obtain an uncertainty score $u_i$ for $x_i$ with $k$-shot ICL.
8: **end for**
9: Determine hard set $\mathcal{U}_h$ given scores $\{u_i\}$ and threshold $\theta$.
10: Perform $k$means clustering of $\mathcal{U}_h$ into $B$ clusters with centroids $\{\mu_j\}_{j=1}^B$.
11: **for** $j = 1, \ldots, B$ **do**
12:     $x_i^* = \arg\min_{x_i \in \mathcal{U}_h} \|\mu_j - x_i\|$.
13:     Add the selected $x_i^*$ to $\mathcal{L} = \mathcal{L} \cup \{x_i^*\}$ and remove it from $\mathcal{U}_h = \mathcal{U}_h \setminus \{x_i^*\}$.
14: **end for**

---

## B.2 ADAICL ALGORITHMS

Algorithm 3 summarizes the overall ADAICL algorithm, and Algorithm 4 summarizes the overall ADAICL+ algorithm. Their key differences are the following. First, ADAICL solves a MAXCOVER problem (Line 14), while ADAICL solves a weighted MAXCOVER problem (Line 14). Second, ADAICL+ performs a predefined number of iterations $T$ (Line 7), sampling a fixed number of examples $B_{cur} = \frac{B}{T}$ (Line 14) per iteration. ADAICL is repeated until termination (Line 7), where the number of selected examples per iteration is determined by the MAXCOVER (Line 14) solution.

---

**Algorithm 3** ADAICL Algorithm.

---

1: **Input:** Model $M$, Unlabeled Set $\mathcal{U}$, Budget $B$, Similarity Space $\mathcal{S}$ for $k$-NN Retriever.
2: **Optional:** Initial set $\mathcal{L}_0$, else $\mathcal{L}_0 = \emptyset$.
3: **Hyperparameters:** threshold $\theta$, number of neighbors $m$.
4: **Output:** Annotated Set $\mathcal{L}$.

5: $B_{cur} = 0, \mathcal{L} = \mathcal{L}_0$.
6: Create global graph $\mathcal{G}_m$.
7: **while** $B_{cur} < B$ **do**
8:     **for** $x_i \in \mathcal{U}$ **do**
9:         Retrieve (at most) $k$ examples from $\mathcal{L}$ based on similarity $\mathcal{S}$.
10:         Use model $M$ to obtain an uncertainty score $u_i$ for $x_i$ with $k$-shot ICL.
11:     **end for**
12:     Determine hard set $\mathcal{U}_h$ given scores $\{u_i\}_{i=1}^N$ and threshold $\theta$.
13:     Create sets $S_i$ given $\mathcal{U}_h$ and $\mathcal{G}_m$.
14:     $\{x_i^*\}_{i=1}^{B^*} = \text{Greedy-MAXCOVER}\left(\mathcal{U}_h, \{S_i\}, B - B_{cur}\right)$.
15:     Add the selected $\{x_i^*\}_{i=1}^{B^*}$ to $\mathcal{L} = \mathcal{L} \cup \{x_i^*\}_{i=1}^{B^*}$ and remove them from $\mathcal{U} = \mathcal{U} \setminus \{x_i^*\}_{i=1}^{B^*}$.
16:     $B_{cur} = B_{cur} + B^*$.
17: **end while**

---

---

**Algorithm 4** ADAICL+ Algorithm.

---

1: **Input:** Model $M$, Unlabeled Set $\mathcal{U}$, Budget $B$, Similarity Space $\mathcal{S}$ for $k$-NN Retriever.
2: **Optional:** Initial set $\mathcal{L}_0$, else $\mathcal{L}_0 = \emptyset$.
3: **Hyperparameters:** threshold $\theta$, number of neighbors $m$, iterations $T$.
4: **Output:** Annotated Set $\mathcal{L}$.

5: $B_{cur} = \frac{B}{T}, \mathcal{L} = \mathcal{L}_0$.
6: Create global graph $\mathcal{G}_m$.
7: **for** $t \in [1, T]$ **do**
8:     **for** $x_i \in \mathcal{U}$ **do**
9:         Retrieve (at most) $k$ examples from $\mathcal{L}$ based on similarity $\mathcal{S}$.
10:         Use model $M$ to obtain an uncertainty score $u_i$ for $x_i$ with $k$-shot ICL.
11:     **end for**
12:     Determine hard set $\mathcal{U}_h$ given scores $\{u_i\}_{i=1}^N$ and threshold $\theta$.
13:     Create sets $S_i$ given $\mathcal{U}_h$ and $\mathcal{G}_m$.
14:     $\{x_i^*\}_{i=1}^{B_{cur}} = $ Greedy-weighted-MAXCOVER $\left(\mathcal{U}_h, \{S_i\}, B_{cur}\right)$.
15:     Add the selected $\{x_i^*\}_{i=1}^{B_{cur}}$ to $\mathcal{L} = \mathcal{L} \cup \{x_i^*\}_{i=1}^{B_{cur}}$ and remove them from $\mathcal{U} = \mathcal{U} \setminus \{x_i^*\}_{i=1}^{B_{cur}}$.
16: **end for**

---

We provide an example of ADAICL+'s advantage in Figure 9. In this example, ADAICL+'s re-weighting schema scores Sets A and B higher than Set C, which contains only a single hard example. On the contrary, if Set A is selected by ADAICL, all the examples of Set B would be marked as covered resulting in a zero total score. The next best scoring set would be Set C, which does not effectively represent the hardness of the examples. In ADAICL+, Set B is scored higher than Set C due to its new weight $w = 0.2$.

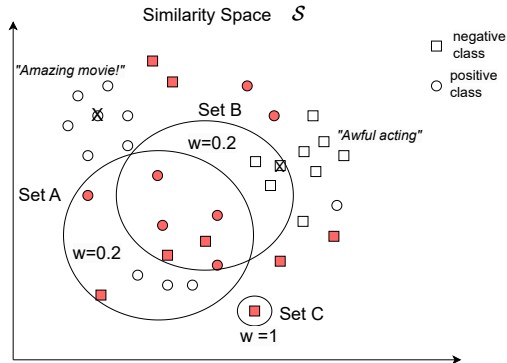

Figure 9: A beneficial case for ADAICL+.

## B.3 SET CONSTRUCTION

In this work, we represent the region $S_i$ around each example $x_i$ as its egonet. Initially, we build a global graph $\mathcal{G}_m$ as the $m$-nearest neighbors graph. The nearest neighbors are obtained with similarity metrics based on the space $\mathcal{S}$, i.e., via cosine similarity. For example, we can have the following edge sets for two nodes $v_1$ and $v_2$ with $m = 4$: $\{v_1 \to v_2, \ v_1 \to v_3, \ v_1 \to v_4, \ v_1 \to v_5\}$, and $\{v_2 \to v_3, \ v_2 \to v_6, \ v_2 \to v_7, \ v_2 \to v_1\}$.

Deriving sets $S_i$ from the global graph $\mathcal{G}_m$ depends on the LLM's feedback as we are interested in hard examples $x_i \in \mathcal{U}_h$ for the model. Thus, we *color* each node as a hard or an easy node based on whether they belong to $\mathcal{U}_h$. For example, we can have $\mathcal{U}_h = \{v_1, v_2, v_3, v_4\}$ and $\{v_5, v_6, v_7\} \notin U_h$ for the case above.

For every hard node $v \in \mathcal{U}_h$, we construct its 1-hop egonet. We consider edges that *direct towards* $v$ from other hard nodes $v' \in \mathcal{U}_h$. This ensures that representative examples $x_i$, that are likely to be retrieved during ICL inference, have denser egonets. For example, if we have $v_2 \to v_1, v_3 \to v_1$ (we exclude links from easy nodes, e.g., $v_5 \to v_1$), we obtain $S_1 = $ egonet$(v_1) = \{v_2, v_3\}$. Similarly, we obtain egonets for other nodes, e.g., $S_2 = $ egonet$(v_2) = \{v_1, v_4\}$, while some nodes might have empty egonets if only easy nodes directs towards them, e.g., $S_3 = $ egonet$(v_3) = \emptyset$. We experiment with both 1-hop and 2-hop set constructions. In the latter case, each node $v$ is represented by its egonet along with union of the egonets of its neighbors, e.g., $S_1^{(2)} = $ ego$(v_1) \cup \{$ego$(v') : v' \in$ ego$(v_1)\} = \{v_2, v_3\} \cup$ ego$(v_2) \cup$ ego$(v_3) = \{v_2, v_3, v_4\}$.

One hyperparameter that controls the quality of the generated egonets is $m$, which is used during the construction of graph $\mathcal{G}_m$. In order to determine $m$, we employ a heuristic rule based on the desired maximum iterations $\hat{T}$ until the budget $B$ is exhausted, as well as the minimum number of

hard examples $N_{\hat{\theta}}$ to be covered at each iteration, where $N_{\hat{\theta}} = \lfloor \hat{\theta} N_\theta \rfloor$ and $\hat{\theta} \in [0, 1]$ is a hyper-parameter with default value $\hat{\theta} = 0.5$. Assuming the graph has reciprocal edges, each node has approximately $\theta m$ and $\theta^2 m^2$ hard examples as neighbors for 1-hop and 2-hop sets, respectively. If at each iteration we annotate $\frac{B}{\hat{T}}$ examples, and we wish to cover at least $N_{\hat{\theta}}$ hard examples, we need to satisfy $N_{\hat{\theta}} \leq \frac{B}{\hat{T}} \theta m$ (for 1-hop sets) and $N_{\hat{\theta}} \leq \frac{B}{\hat{T}} \theta^2 m^2$ (for 2-hop sets). Thus, the heuristic-based rule is given by

$$\begin{cases} \frac{\hat{T} N_{\hat{\theta}}}{\theta B} \leq m \leq \frac{\hat{T} N_\theta}{\theta B} & \text{for 1-hop sets,} \\ \frac{\hat{T} N_{\hat{\theta}}}{\theta^2 B} \leq m^2 \leq \frac{\hat{T} N_\theta}{\theta^2 B} & \text{for 2-hop sets,} \end{cases} \tag{6}$$

which is adjustable to the portion of the examples that we account as hard ones (the right hand side is derived due to constraint of maximum iterations $\hat{T}$). Moreover, instead of having a $m$-nn graph $\mathcal{G}_m$, we experiment with a threshold-based $\delta$-graph $\mathcal{G}_\delta$, where we set the threshold accordingly.

## C  PIPELINE OF COMPARED METHODS

We clarify on the pipeline of all compared methods. All methods have a "Selection Phase", which selects which examples to annotate, and an "Inference Phase", which is the $k$-shot retrieval-based ICL and is the same for all. We use $k$-shot retrieval-based inference as it is shown to be more effective for ICL (Margatina et al., 2023). We provide the comparison Table 3, where compared methods differ during the "Selection Phase". As it is shown, all methods use the same retriever during inference and have the same computation cost. During selection, model-based methods (Votek, AdaICL) have a higher cost, but this cost is only needed before inference/deployment. We have performed a time analysis in Appendix E.6 for the example selection process: With the downsampled version of data, AdaICL outperforms Random without involving many LLM queries. During inference, all methods (Random, Votek, AdaICL) have the same cost.

Table 3: Pipeline and complexity of compared methods.

| Method | Selection | | Inference |
|---|---|---|---|
| | How $\mathcal{L}$ is constructed, $B$ examples | Complexity | Using examples from $\mathcal{L}$ for ICL (same for all) |
| Random | Random | Zero-cost | $k$-shot ($k \ll B$) retrieval |
| $K$means | Clustering | Independent of the LLM | $k$-shot ($k \ll B$) retrieval |
| Hardest | Uncertainty | Depends on LLM's complexity | $k$-shot ($k \ll B$) retrieval |
| Vote$k$ | Vote$k$ | Depends on LLM's complexity | $k$-shot ($k \ll B$) retrieval |
| ADAICL | ADAICL | Depends on LLM's complexity | $k$-shot ($k \ll B$) retrieval |

## D  EXPERIMENTAL SETTING DETAILS

### D.1  DATASETS

We performed empirical evaluation with nine NLP datasets that cover well-studied tasks, such as topic classification (AGNews (Zhang et al., 2015), TREC (Hovy et al., 2001)), sentiment analysis (SST2 (Socher et al., 2013), Amazon (McAuley & Leskovec, 2013)), natural language inference (RTE (Bentivogli et al., 2009), MRPC (Dolan et al., 2004), MNLI (Williams et al., 2018)), text summarization (XSUM (Narayan et al.)) and math reasoning (GSM8K (Cobbe et al., 2021)). We provide examples of these datasets in Table 4, which we access via Hugging Face package (Lhoest et al., 2021).

Each dataset contains official train/dev/test splits. We follow Vote$k$ and sample 256 examples randomly from the test set (if it is publicly available, otherwise from the dev set) as test data. For the train data, we remove the annotations before our active learning setup. As it is infeasible to evaluate the LLM's feedback on all instances due to computational constraints, e.g., Amazon dataset has more than 1 million instances, we randomly subsample 3,000 instances, which we cluster into 310 groups, and we select the 310 examples closest to the centroids as candidate examples for annotation. We repeat the above processes for both the train and test sets three times with different seeds and report mean and standard deviation results. In transductive settings, we only consider the test data for annotation. In this case, we evaluate performance on all test examples, but we also exclude retrieving examples that lead to self-label leakage issues.

Table 4: Dataset examples. $< S_1 >$ denotes the input sequences.

| Dataset | Task | Example $x$ | Labels/Annotations $y$ |
|---------|------|-------------|------------------------|
| AGNews | Topic Classification | $< S_1 >$: "Amazon Updates Web Services Tools, Adds Alexa Access The Amazon Web Services (AWS) division of online retail giant Amazon.com yesterday released Amazon E-Commerce Service 4.0 and the beta version of Alexa Web Information Service." | World, Sport, Business, Sci-Tech |
| TREC | Answer Type Classification | $< S_1 >$: "What is the date of Boxing Day?" | Abbreviation, Entity, Description, Human, Location, Numeric |
| SST2 | Sentiment Analysis | $< S_1 >$: "covers this territory with wit and originality , suggesting that with his fourth feature" | Positive, Negative |
| Amazon | Sentiment Analysis | $< S_{1a} >$:"Very Not Worth Your Time", $< S_{1b} >$:"The book was written very horribly. I would never in my life recommend such a book..." | Positive, Negative |
| RTE | Natural Language Inference | $< S_1 >$:"In a bowl, whisk together the eggs and sugar until completely blended and frothy.", $< S_2 >$:"In a bowl, whisk together the egg, sugar and vanilla until light in color." | Entailment, Not Entailment |
| MRPC | Paraphrase Detection | $< S_1 >$:"He said the foodservice pie business doesn't fit the company's long-term growth strategy.", $< S_2 >$:"The foodservice pie business does not fit our long-term growth strategy." | Equivalent, Not Equivalent |
| MNLI | Natural Language Inference | $< S_1 >$:"The new rights are nice enough", $< S_2 >$: "Everyone really likes the newest benefits" | Entailment, Neutral, Contradiction |
| XSUM | Summarization | $< S_1 >$:"The 3kg (6.6lb) dog is set to become part of a search-and-rescue team used for disasters such as earthquakes. Its small size means it will be able to squeeze into places too narrow for dogs such as German Shepherds. Chihuahuas, named after a Mexican state, are one of the the smallest breeds of dog. "It's quite rare for us to have a chihuahua work as a police dog (said a police spokeswoman in Nara, western Japan). We would like it to work hard by taking advantage of its small size. Momo, aged seven, will begin work in January." | "A chihuahua named Momo (Peach) has passed the exam to become a dog in the police force in western Japan, in what seems to be a first." |
| GSM8K | Math Reasoning | $< S_1 >$:"James writes a 3-page letter to 2 different friends twice a week. How many pages does he write a year?" | "He writes each friend 3*2=6 pages a week So he writes 6*2=12 pages every week That means he writes 12*52=624 pages a year. Thus, the answer is 624." |

## D.2 PROMPTS

As a design choice of the input prompts, we slightly modify the templates proposed by Gao et al. (2021) to transform them as a continuation task. We find that these are more challenging prompts for the large LMs, which we present in Table 5 (top). In Section E.4, we also experiment with alternative prompt templates similar to Su et al. (2022), as shown in Table 5 (bottom).

## D.3 CONFIGURATIONS

As summarized in Figure 2, the design space includes the unlabeled set $\mathcal{U}$, the number of ICL examples $k$, the similarity space $\mathcal{S}$, the budget $B$, and the LLM $M$. We use the default hyperparameters of the Transformers library (Wolf et al., 2020) for each LLM. We obtain the initial pool of annotated examples $\mathcal{L}_0$ via $k$means so that we reduce randomness. We summarize the experimental configurations in Table 6.

Table 5: Prompt templates for the ICL demonstrations.

| Task | Template | Continuation (label word) |
|---|---|---|
| | **Default** | |
| AGNews | Content: $< S_1 > \backslash n$ | World, Sport, Business, Sci-Tech |
| TREC | Content: $< S_1 > \backslash n$ | Abbreviation, Entity, Description, Human, Location, Numeric |
| SST2 | $< S_1 >$. It was | great, terrible |
| Amazon | $< S_{1a} >< S_{1b} >$. It was | great, terrible |
| RTE | $< S_1 >$? [MASK], $< S_2 >$ | [MASK]: Yes, [MASK]: No |
| MRPC | $< S_1 >$? [MASK], $< S_2 >$ | [MASK]: Yes, [MASK]: No |
| MNLI | $< S_1 >$? [MASK], $< S_2 >$ | [MASK]: Yes, [MASK]: Maybe, [MASK]: No |
| | **Alternative** | |
| AGNews | Content: $< S_1 >$ Topic: | World, Sport, Business, Sci-Tech |
| TREC | Content: $< S_1 >$ Answer Type: | Abbreviation, Entity, Description, Human, Location, Numeric |
| SST2 | Content: $< S_1 >$ Sentiment: | Positive, Negative |
| Amazon | Title: $< S_{1a} >$ Review: $< S_{1b} >$ Sentiment: | Positive, Negative |
| RTE | $< S_1 >$. Question: $< S_2 >$. True or False? Answer: | True, False |
| MRPC | Are the following sentences equivalent or not equivalent? $< S_1 > \backslash n < S_2 >$ | equivalent, not equivalent |
| MNLI | $< S_1 >$. Based on that information, is the claim $< S_2 >$ True, False, or Inconclusive? Answer: | True, Inconclusive, False |

Table 6: Experimental setting configurations.

| Setting | Models $M$ | Train/Test $\mathcal{U}$ | Budget $B$ | $k$-shot | Retriever, $\mathcal{S}$ | Init. |
|---|---|---|---|---|---|---|
| **Main** | | | | | | |
| Figure 4 | GPTJ, GPT-Neo | Inductive | 20 | 5 | SBERT | $|\mathcal{L}_0| = 10$ |
| XSUM | Falcon-40B, LLaMa-65B | Transductive | 10 | Context-limit | SBERT | Zero-shot |
| GSM8K | Falcon-40B, LLaMa-65B | Transductive | 20 | 5 | BERT,SBERT | Zero-shot |
| Table 1 | GPT-J, MPT | Transductive | 5,10 | 5,10 | SBERT | Zero-shot |
| Table 2 | GPT-Neo | Inductive | 20 | 5 | SBERT, RoBERTa, BERT | $|\mathcal{L}_0| = 10$ |
| Figure 5 | GPT-Neo | Transductive | 0-45 | Context-limit | SBERT | Zero-shot |
| Figure 6 | GPT-J, MPT, Falcon, LLaMa (6-7B) | Transductive | 0-20 | Context-limit | SBERT | Zero-shot |
| Figure 7 | GPT-Neo | Transductive | 0-45 | Context-limit | SBERT | Zero-shot |
| Appendices E, F | GPT-J, GPT-Neo | Inductive | 20 | 5 | SBERT | $|\mathcal{L}_0| = 10$ |

Context-limit means that we retrieve as many few-shot examples as the input token-length limit allows. For example, XSUM has long sequences, where we usually have 3-shot examples, while for TREC we can use as many as 80-shot examples.

# E  FURTHER RESULTS

## E.1  BASE RESULTS (FULL)

Tables 7 and 8 give the full results of Figure 4 for GPT-J and GPT-Neo, accordingly. ADAICL performs the best over all tasks, while the second-best method is ADAICL-base.

Table 7: Performance comparison for GPT-J (6B).

| | Topic Classification | | Sentiment Analysis | | Natural Language Inference | | |
|---|---|---|---|---|---|---|---|
| | AGNews | TREC | SST2 | Amazon | RTE | MRPC | MNLI |
| Random | $68.87_{\pm 5.39}$ | $49.34_{\pm 3.19}$ | $81.63_{\pm 0.30}$ | $87.89_{\pm 1.77}$ | $52.86_{\pm 2.41}$ | $69.01_{\pm 4.61}$ | $39.58_{\pm 3.98}$ |
| Fast-vote$k$ | $73.69_{\pm 2.39}$ | $49.61_{\pm 4.43}$ | $78.99_{\pm 4.53}$ | $89.58_{\pm 0.80}$ | $53.00_{\pm 0.49}$ | $68.23_{\pm 2.89}$ | $39.97_{\pm 3.98}$ |
| Vote$k$ | $72.26_{\pm 1.27}$ | $45.83_{\pm 1.75}$ | $80.45_{\pm 1.47}$ | $85.80_{\pm 3.80}$ | $54.16_{\pm 2.30}$ | $68.10_{\pm 2.75}$ | $39.72_{\pm 2.07}$ |
| Hardest | $72.13_{\pm 2.12}$ | $35.93_{\pm 5.53}$ | $82.67_{\pm 1.64}$ | $87.36_{\pm 0.66}$ | $55.33_{\pm 2.25}$ | $66.80_{\pm 5.25}$ | $38.80_{\pm 1.11}$ |
| ADAICL-base | $73.56_{\pm 2.96}$ | $50.64_{\pm 9.11}$ | $84.11_{\pm 3.25}$ | $91.01_{\pm 1.77}$ | $52.73_{\pm 2.21}$ | $66.53_{\pm 4.78}$ | $38.66_{\pm 4.50}$ |
| Best (Avg.) | ADAICL-base (62.10) | | ADAICL-base (87.56) | | Vote$k$ (53.99) | | |
| ADAICL | $76.89_{\pm 3.01}$ | $51.95_{\pm 8.43}$ | $82.81_{\pm 1.39}$ | $90.49_{\pm 1.57}$ | $56.90_{\pm 1.75}$ | $70.17_{\pm 1.72}$ | $40.36_{\pm 1.75}$ |
| ADAICL+ | $77.08_{\pm 1.11}$ | $53.38_{\pm 5.10}$ | $84.24_{\pm 1.32}$ | $92.45_{\pm 1.50}$ | $55.07_{\pm 0.85}$ | $68.49_{\pm 0.97}$ | $36.58_{\pm 1.12}$ |
| Best (Avg.) | ADAICL+ (65.23) | | ADAICL+ (88.35) | | ADAICL (55.81) | | |
| $\Delta$-Gain (Absolute) | +3.13 | | +0.79 | | +1.82 | | |

## E.2  MULTI-STEP RESULTS

Figure 10 shows the full results for multi-step active learning with different algorithms. ADAICL outperforms all other model-based methods (ADAICL-base, Vote$k$, Patron, Hardest) in all tasks, almost at every step. ADAICL-base is the second-best at topic classification, while Patron is the second-best at sentiment analysis. For natural language inference, random selection outperforms most other methods, but although the model's feedback does not necessarily help the taks, ADAICL outperforms all other model-based methods.

Table 8: Performance comparison for GPT-Neo (1.3B).

| | Topic Classification | | Sentiment Analysis | | Natural Language Inference | | |
| | AGNews | TREC | SST2 | Amazon | RTE | MRPC | MNLI |
|---|---|---|---|---|---|---|---|
| Random | $59.47_{\pm 8.54}$ | $54.68_{\pm 1.68}$ | $68.48_{\pm 1.87}$ | $73.95_{\pm 2.03}$ | $48.30_{\pm 1.30}$ | $64.48_{\pm 7.67}$ | $40.99_{\pm 0.97}$ |
| Fast-vote$k$ | $62.23_{\pm 3.89}$ | $46.48_{\pm 3.04}$ | $69.78_{\pm 8.34}$ | $69.39_{\pm 0.98}$ | $50.64_{\pm 1.02}$ | $64.19_{\pm 0.97}$ | $38.40_{\pm 0.92}$ |
| Vote$k$ | $62.77_{\pm 4.82}$ | $53.12_{\pm 4.07}$ | $73.69_{\pm 9.05}$ | $75.13_{\pm 0.98}$ | $49.99_{\pm 0.32}$ | $67.44_{\pm 2.96}$ | $39.18_{\pm 1.60}$ |
| Hardest | $65.10_{\pm 2.43}$ | $49.34_{\pm 2.17}$ | $71.48_{\pm 5.32}$ | $75.00_{\pm 2.49}$ | $52.86_{\pm 0.80}$ | $61.84_{\pm 4.79}$ | $37.49_{\pm 1.77}$ |
| ADAICL-base | $70.17_{\pm 1.84}$ | $48.24_{\pm 0.98}$ | $77.86_{\pm 1.02}$ | $75.77_{\pm 3.62}$ | $53.77_{\pm 0.73}$ | $64.71_{\pm 7.39}$ | $39.71_{\pm 1.03}$ |
| Best (Avg.) | ADAICL-base (59.21) | | ADAICL-base (76.82) | | Vote$k$ (52.20) | | |
| ADAICL | $70.95_{\pm 1.87}$ | $55.33_{\pm 2.57}$ | $79.68_{\pm 1.77}$ | $77.73_{\pm 2.23}$ | $53.12_{\pm 1.59}$ | $67.05_{\pm 8.10}$ | $42.96_{\pm 2.92}$ |
| ADAICL+ | $69.39_{\pm 1.35}$ | $59.89_{\pm 2.07}$ | $79.03_{\pm 2.47}$ | $77.08_{\pm 1.50}$ | $51.16_{\pm 1.39}$ | $65.69_{\pm 8.92}$ | $40.49_{\pm 2.04}$ |
| Best (Avg.) | ADAICL+ (64.64) | | ADAICL (78.71) | | ADAICL (54.38) | | |
| $\Delta$-Gain (Absolute) | +5.53 | | +1.99 | | +2.28 | | |

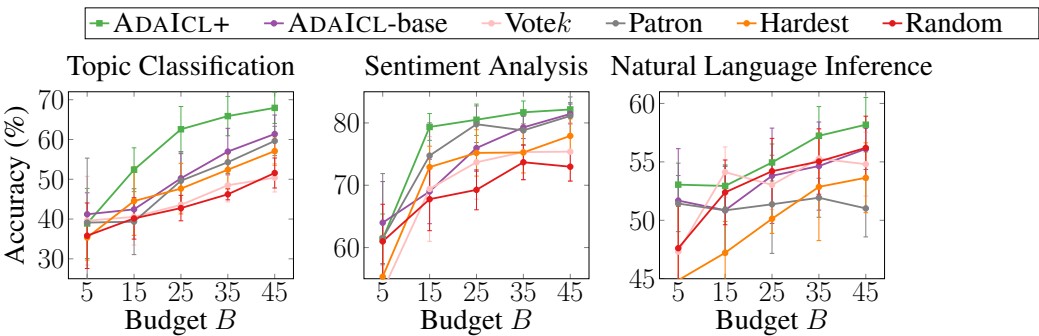

Figure 10: Multi-step results with GPT-Neo.

### E.3 RETRIEVER ABLATION

A benefit of the $k$-NN retriever is that it can determine the order of the input few-shot examples by semantic similarity scores. In general, we place demonstrations with higher similarity closer to the test instance. Table 9 gathers results for different retrievers as well as when we randomly re-order the input demonstrations. ADAICL is the most robust method and outperforms other baselines regardless of the choice of the retriever.

### E.4 PROMPT TEMPLATE ABLATION

Table 10 reports results when we use alternative ICL prompt templates (Table 5) for the input examples. ADAICL is robust to the design of the prompt templates, where it outperforms other baselines in most datasets.

Table 9: Performance comparison across different retrieval and semantic similarity configurations.

| Retriever, $\mathcal{S} \rightarrow$ | SBERT-all-mpnet-base | | | RoBERTa-nli-large-mean-tokens | | | BERT-nli-large-cls-pool | | | Avg. |
| | TREC | SST2 | Amazon | TREC | SST2 | Amazon | TREC | SST2 | Amazon | |
|---|---|---|---|---|---|---|---|---|---|---|
| Random | $54.68_{\pm 1.68}$ | $68.48_{\pm 1.87}$ | $73.95_{\pm 2.03}$ | $37.23_{\pm 2.30}$ | $74.21_{\pm 3.50}$ | $84.46_{\pm 3.21}$ | $34.75_{\pm 2.41}$ | $72.65_{\pm 5.82}$ | $80.20_{\pm 3.34}$ | 64.51 |
| Vote$k$ | $54.81_{\pm 0.49}$ | $73.69_{\pm 9.05}$ | $75.13_{\pm 0.98}$ | $37.77_{\pm 4.65}$ | $76.16_{\pm 2.23}$ | $84.11_{\pm 1.28}$ | $42.43_{\pm 3.34}$ | $80.85_{\pm 2.09}$ | $83.59_{\pm 1.77}$ | 67.61 |
| ADAICL-base | $48.24_{\pm 0.98}$ | $77.86_{\pm 1.02}$ | $75.77_{\pm 3.63}$ | $38.12_{\pm 5.74}$ | $78.12_{\pm 5.30}$ | $\underline{85.93}_{\pm 2.30}$ | $38.15_{\pm 3.10}$ | $78.64_{\pm 2.78}$ | $\underline{85.80}_{\pm 1.75}$ | 67.40 |
| ADAICL | $\underline{55.33}_{\pm 2.57}$ | $\underline{79.68}_{\pm 2.47}$ | $\underline{77.73}_{\pm 2.23}$ | $\underline{39.06}_{\pm 3.37}$ | $\underline{81.11}_{\pm 1.50}$ | $85.15_{\pm 0.55}$ | $44.06_{\pm 2.49}$ | $\underline{80.85}_{\pm 2.83}$ | $84.65_{\pm 3.52}$ | **69.74** |
| *random reorder* | | | | | | | | | | |
| Random | $47.39_{\pm 2.89}$ | $66.57_{\pm 3.86}$ | $76.42_{\pm 2.57}$ | $31.87_{\pm 4.45}$ | $69.74_{\pm 5.84}$ | $81.24_{\pm 2.72}$ | $40.34_{\pm 1.58}$ | $76.55_{\pm 4.06}$ | $79.54_{\pm 2.64}$ | 63.29 |
| Vote$k$ | $45.43_{\pm 2.94}$ | $72.00_{\pm 3.73}$ | $73.04_{\pm 1.77}$ | $37.62_{\pm 5.15}$ | $75.91_{\pm 5.58}$ | $82.03_{\pm 0.84}$ | $39.57_{\pm 3.19}$ | $79.29_{\pm 2.55}$ | $81.24_{\pm 1.43}$ | 65.13 |
| ADAICL-base | $50.64_{\pm 5.43}$ | $74.99_{\pm 6.37}$ | $74.86_{\pm 4.65}$ | $36.56_{\pm 2.98}$ | $75.77_{\pm 0.63}$ | $84.89_{\pm 2.02}$ | $\underline{45.56}_{\pm 3.01}$ | $79.55_{\pm 0.18}$ | $85.41_{\pm 0.91}$ | 67.58 |
| ADAICL | $52.22_{\pm 5.19}$ | $78.12_{\pm 4.71}$ | $75.64_{\pm 3.51}$ | $38.93_{\pm 0.48}$ | $76.68_{\pm 2.71}$ | $84.11_{\pm 1.43}$ | $44.00_{\pm 1.63}$ | $78.51_{\pm 4.52}$ | $85.28_{\pm 4.65}$ | **68.16** |

Table 10: Prompt template ablation study.

| | GPT-Neo | | | | GPT-J | | |
| | AGNews | TREC | SST2 | Amazon | RTE | MRPC | MNLI |
|---|---|---|---|---|---|---|---|
| **Default Prompts** | | | | | | | |
| Random | $59.47_{\pm8.54}$ | $54.68_{\pm1.68}$ | $68.48_{\pm1.87}$ | $73.95_{\pm2.03}$ | $52.86_{\pm2.41}$ | $69.01_{\pm4.61}$ | $39.58_{\pm3.98}$ |
| Vote$k$ | $62.77_{\pm4.82}$ | $53.12_{\pm4.07}$ | $73.69_{\pm9.05}$ | $75.13_{\pm0.98}$ | $54.16_{\pm2.30}$ | $68.10_{\pm2.75}$ | $39.72_{\pm2.07}$ |
| ADAICL | $70.95_{\pm1.87}$ | $55.33_{\pm2.57}$ | $79.68_{\pm1.77}$ | $77.73_{\pm2.23}$ | $56.90_{\pm1.75}$ | $70.17_{\pm1.72}$ | $40.36_{\pm1.75}$ |
| **Alternative Prompts** | | | | | | | |
| Random | $73.69_{\pm1.21}$ | $51.76_{\pm4.55}$ | $59.89_{\pm3.98}$ | $73.82_{\pm3.35}$ | $56.41_{\pm2.13}$ | $56.37_{\pm3.72}$ | $38.93_{\pm1.18}$ |
| Vote$k$ | $72.78_{\pm2.12}$ | $50.38_{\pm5.90}$ | $64.84_{\pm2.92}$ | $73.43_{\pm2.23}$ | $56.38_{\pm2.70}$ | $51.95_{\pm2.53}$ | $40.49_{\pm2.05}$ |
| ADAICL | $76.95_{\pm1.27}$ | $54.94_{\pm1.43}$ | $65.88_{\pm4.58}$ | $75.64_{\pm1.29}$ | $56.37_{\pm1.29}$ | $59.22_{\pm2.39}$ | $35.40_{\pm1.31}$ |

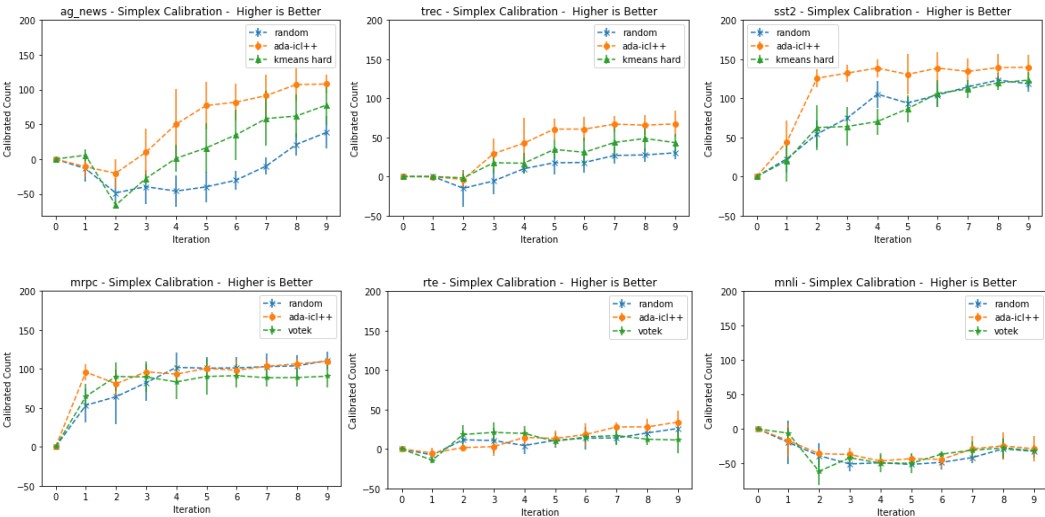

Figure 11: Calibration analysis via simplicies where we compare ADAICL against the random selection algorithm and the best performing baseline for each dataset.

### E.5 CALIBRATION ANALYSIS VIA SIMPLICES

Calibration has recently also been studied from the perspective of simplices (Heese et al., 2023). Abstractly, a simplex is the generalization of the notion of a triangle/ tetrahedron to arbitrary dimensions. Here, we present a simplified study of calibration via the lens of simplices, where we test if the predicted label of the test instance lies within the simplex given by the retrieved examples *having the same label*. In more detail, given a prediction $y_{test} = \boldsymbol{y}$ for test instance $x_{test}$, we first obtain the subset of retrieved examples for $x_{test}$, i.e. $\mathcal{X}_{subset} \subseteq \mathcal{X}_{retrieved} \subset B$ which share the same label $\boldsymbol{y}$. We then construct a simplex using the SBERT embeddings of $\mathcal{X}_{subset}$, run PCA to obtain low dimensional embeddings and test to see if the low dimensional embedding of $x_{test}$ lies within the above constructed simplex.

To take into consideration the effect of overconfident but wrong predictions, for a given dataset, we subtract the total counts of the cases when $\boldsymbol{y} \neq \boldsymbol{y}_{true}$ from the cases where $\boldsymbol{y} = \boldsymbol{y}_{true}$ for all $x_{test}$ in the dataset. Results are presented in the plots in Figure 11 where we compare ADAICL against random and the best performing baseline on that dataset. For AGNews, TREC, and SST2, the label information of the retrieved examples is important and ADAICL leads to the best calibration in these cases. For RTE, MRPC, and MNLI, retrieving examples of the same label is not crucial for performance and thus, all methods behave similarly with respect to simplex calibration.

### E.6 SELECTION TIME COMPLEXITY

In Table 11, we compare competing approaches based on their computation time during their selection process (during downstream inference, their time complexity is the same). Random selection has zero cost. Vote$k$ and ADAICL ($T = 1$) have the same cost, while the cost doubles for ADAICL ($T = 2$). Nevertheless, hyper-parameter $T$ for ADAICL can be tuned depending on the desired runtime of the selection process.

Table 11: Time complexity analysis with 5-shot ICL for different selection processes over 300 examples on a GeForce RTX 3090 (24GB GPU).

| | Embedding Computation (SBERT) | Model Uncertainty Estimation (GPT-Neo) |
|---|---|---|
| **Amazon** | | |
| Random | 0 secs | 0 secs |
| Vote$k$ | $\approx$1 secs | 1 min & 51 secs |
| ADAICL ($T = 1$) | $\approx$1 secs | 1 min & 51 secs |
| ADAICL ($T = 2$) | $\approx$1 secs | $\approx$ 3 mins & 42 secs |
| **AGNews** | | |
| Random | 0 secs | 0 secs |
| Vote$k$ | $\approx$0.5 secs | 3 mins & 48 secs |
| ADAICL ($T = 1$) | $\approx$0.5 secs | 3 mins & 48 secs |
| ADAICL ($T = 2$) | $\approx$0.5 secs | $\approx$ 7 mins & 36 secs |

## F ADAICL ABLATION STUDIES

### F.1 GRAPH ABLATION STUDIES

Table 12: Graph ablation study using GPT-Neo based on hyper-parameters $m$, which controls the number of graph neighbors, and $l$, which controls whether we consider 1-hop or 2-hop sets. The value combinations of $m$ and $l$ are adjusted via Equation 6.

| | AGNews | SST2 | Amazon |
|---|---|---|---|
| Vote$k$ | $62.77_{\pm 4.82}$ | $73.69_{\pm 9.05}$ | $75.13_{\pm 0.98}$ |
| ADAICL | | | |
| $m = 15, l = 1$ | $68.61_{\pm 1.02}$ | $79.42_{\pm 1.28}$ | $77.34_{\pm 2.73}$ |
| $m = 25, l = 1$ | $68.74_{\pm 3.59}$ | $77.60_{\pm 4.20}$ | $76.95_{\pm 3.86}$ |
| *$m = 5, l = 2$ | $70.95_{\pm 1.87}$ | $79.68_{\pm 1.77}$ | $77.73_{\pm 2.23}$ |
| $m = 7, l = 2$ | $69.13_{\pm 1.39}$ | $78.12_{\pm 2.61}$ | $74.99_{\pm 0.84}$ |
| ADAICL+ ($T = 2$) | | | |
| *$m = 15, l = 1$ | $69.39_{\pm 1.35}$ | $79.03_{\pm 2.47}$ | $77.08_{\pm 1.50}$ |
| $m = 25, l = 1$ | $68.87_{\pm 3.26}$ | $77.73_{\pm 2.22}$ | $73.17_{\pm 2.59}$ |
| $m = 5, l = 2$ | $70.43_{\pm 1.60}$ | $77.73_{\pm 1.15}$ | $76.43_{\pm 2.55}$ |
| $m = 7, l = 2$ | $70.60_{\pm 2.89}$ | $76.95_{\pm 4.69}$ | $77.21_{\pm 2.12}$ |

*Denotes the default value.

Table 12 shows an ablation study on our proposed heuristic rule of Equation 6. We select $m$ such that it lies near the boundaries of Equation 6, depending whether we choose $l = 1$ hop sets or $l = 2$ hop sets. As Table 12 shows, our heuristic rule is robust and achieves good results with four different combinations. In some cases, having smaller values of $m$ leads to slightly better results as it excludes neighbors with less semantic similarity.

Table 13: Graph ablation study for ADAICL using GPT-J with different graph construction approaches.

| | AGNews | TREC | SST2 | Amazon | RTE | MRPC | MNLI |
|---|---|---|---|---|---|---|---|
| $m$-nn graph | $77.08_{\pm 1.11}$ | $53.38_{\pm 5.10}$ | $84.24_{\pm 1.32}$ | $92.45_{\pm 1.50}$ | $56.90_{\pm 1.75}$ | $70.17_{\pm 1.72}$ | $40.36_{\pm 1.75}$ |
| $\delta$-graph | $76.17_{\pm 3.45}$ | $50.51_{\pm 4.65}$ | $81.90_{\pm 2.48}$ | $88.80_{\pm 1.57}$ | $56.63_{\pm 3.23}$ | $68.75_{\pm 1.93}$ | $40.62_{\pm 2.08}$ |

Furthermore, we also experiment with using a threshold-based graph ($\delta$-graph) instead of the $m$-nn graph. To determine threshold $\delta$, we compute the cosine similarity between all nodes and set

$\delta$ such as each node has $m$ neighbors *on average* (at the $m$-nn graph each nodes has exactly $m$ neighbors). As Table 13 shows, using the $\delta$-graph performs slightly worse than the $m$-nn graph. We hypothesize that using the $\delta$-graph gives more importance on the semantics of the train distribution (as $\delta$ value is computed based on the similarity scores between all train examples), which may not always generalize well to the test distribution.

## F.2 UNCERTAINTY THRESHOLD

Table 14: Ablation study using GPT-Neo based on hyper-parameter $\theta$, which controls the number of the examples that are considered as hard ones.

|  | TREC | SST2 | Amazon |
|---|---|---|---|
| Vote$k$ | $53.12_{\pm 4.07}$ | $73.69_{\pm 9.05}$ | $75.13_{\pm 0.98}$ |
| ADAICL+ ($\theta = 0.5$) | $59.89_{\pm 2.07}$ | $79.03_{\pm 2.47}$ | $77.08_{\pm 1.50}$ |
| ADAICL+ ($\theta = 0.33$) | $60.28_{\pm 3.13}$ | $78.77_{\pm 2.59}$ | $78.90_{\pm 1.14}$ |

By default, we consider 50% ($\theta = 0.5$) of the examples with the lowest confidence as hard examples. Table 14 shows results when we focus on harder examples by setting $\theta = 0.33$ for ADAICL+. Interestingly, ADAICL+'s performance can be further boosted with careful tuning of the uncertainty threshold. Thus, *automatically* determining which examples are considered as hard examples for the models seems a promising research direction.

## G VISUALIZATION

We illustrate the selection process of ADAICL in Figure 12. Initially, the LLMs perform 0-shot ICL but do not make confident predictions (as the hue color, that represents the model's uncertainty for each example, indicates). Note that different LLMs may consider different examples as hard or easy ones. Then, ADAICL selects 5 representative for 5-shot ICL, which improves the LLMs' understanding of the task and reduces its uncertainty (we observe fewer red nodes and more nodes with greener color).

## H ADAICL LIMITATIONS

We list some of our assumptions that may limit ADAICL if they are not satisfied. First, we assume that we can access the output logits/probabilities from the LLM in order to evaluate its uncertainty; this might not be always be feasible. Second, ADAICL relies on embedding methods to determine semantic diversity. While ADAICL is shown to be robust to different methods, it can still suffer if the semantic space of the test is wildly different from the annotation pool space. Finally, the graph/set construction is a heuristic approach and does not account for cases where adversarial examples are injected into the pool in order to degrade performance.

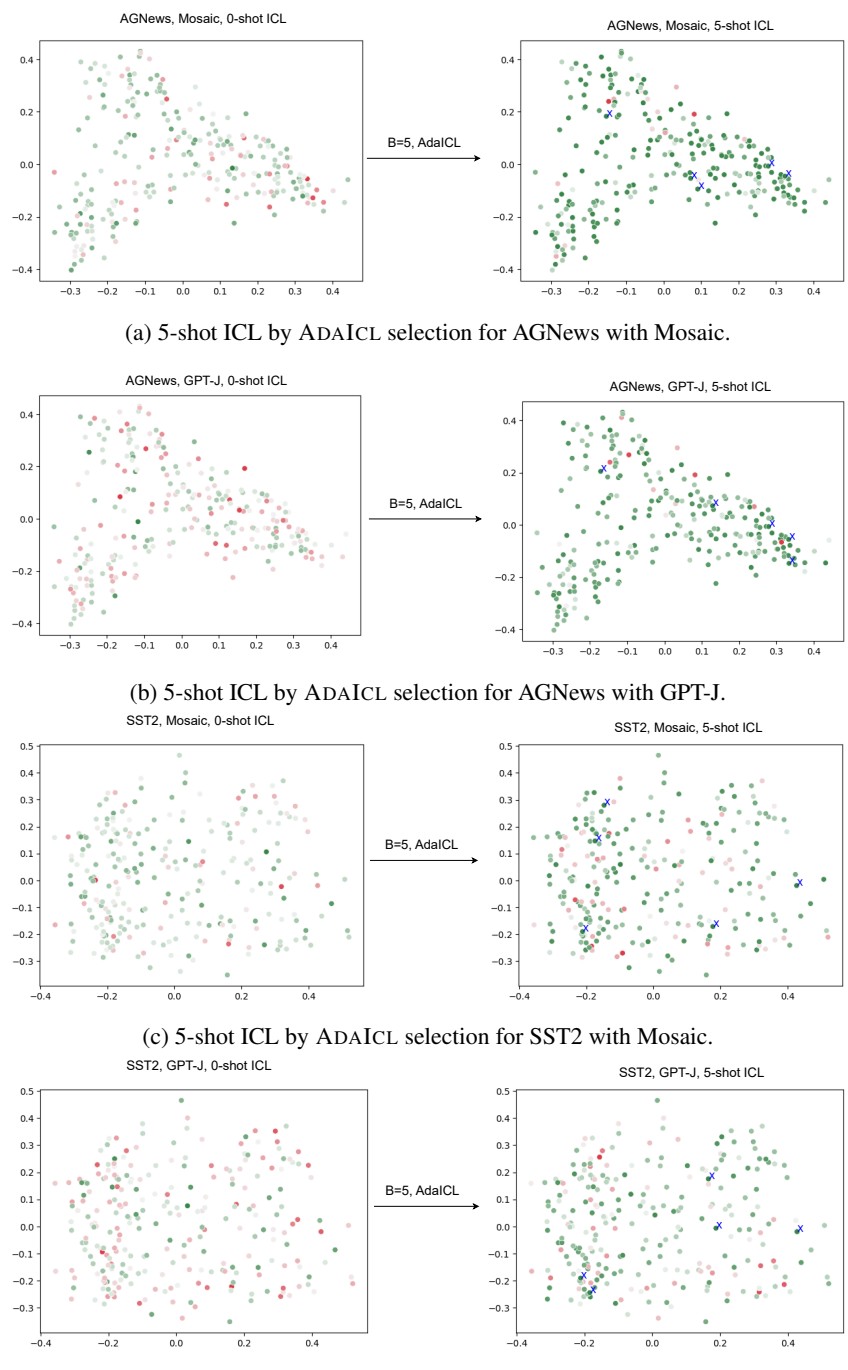

Figure 12: Visualization of the SBERT space (after PCA) and of the examples selected for annotation. The two axes represent the two most important PCA components, which is performed over the SBERT embeddings of the examples. The hue color (green to red) represents the model's uncertainty (confident to uncertain) for each example. The selected examples for annotation are marked with the blue 'X' symbol (better viewed with zooming).

