# OpenReview forum: "Which Examples to Annotate for In-Context Learning? Towards Effective and Efficient Selection"
_ICLR.cc/2024/Conference — Submitted to ICLR 2024_

### Official Review · Reviewer_LBWT · 2023-10-15

**Soundness:** 2 fair
**Presentation:** 2 fair
**Contribution:** 2 fair
**Rating:** 3
**Confidence:** 3

**Summary:**

This paper explores in-context learning (ICL) for Large Language Models (LLMs) where the goal is to adapt the model to new tasks with minimal annotation. It introduces ADAICL, an active learning approach that efficiently selects examples for annotation within a limited budget. ADAICL identifies uncertain examples for the model and uses semantic diversity-based selection. This approach, treated as a maximum coverage problem, dynamically adapts based on the model's feedback. Experiments across datasets and LLMs demonstrate ADAICL's superiority, improving accuracy over state-of-the-art works and being up to 3 times more budget-efficient than random annotation. It also outperformed existing methods with half the number of annotated examples.

**Strengths:**

- The motivation is clear. It is significant to study how to determine which unlabeled instances should be labeled. This is important to reduce the cost of annotations in in-context learning.
- Experimental results are overall great. On a series of tasks, the proposed method can achieve the best performance.

**Weaknesses:**

- The contributions are somewhat overclaimed.
- Technical contributions are not sufficient.
- The writing also should be polished. For the current form, there are a series of unclear justifications.

More details about the above weaknesses can be checked below.

**Questions:**

- It is possible to meet outliers if the method overemphasizes the selection of diverse data. However, in the current form, it is not clear how to address the issue.
- Does the $k$-NN retriever equal the similar retriever in the Vote-$k$ paper?
- This paper claims that "However, these approaches do not consider which examples help the LLM learn new information and may waste resources for annotating examples whose answers are already within the model’s knowledge." I am somewhat confused about this claim. The method Vote-$k$ also uses the feedback of LLMs. Could the paper give more details about this?
- Does the uncertain with respect to one example equal that the LLM can learn it accurately?
- The paper argues that previous work assumes a high-resource setting, where a large set of ICL examples is already annotated. Could the paper provide some detailed examples for better understanding?
- For Section 4.1, could the paper provide more details about how to obtain the probability with respect to the label or demonstrations?
- Compared with previous work such as Vote-$k$, the method proposed by this work is more complex. It introduces a series of hyper-parameters. How to balance them in practice? Also, is there a time advantage of the proposed method over baselines?
- What is the definition of "egonet" in this paper?
- For Figure 5, could the paper supplement the comparison between all methods not just the best baseline?

---

> ### Author Response · Authors · 2023-11-15
> **Response to Reviewer LBWT (1/2)**
>
> We thank the Reviewer for their appreciation of the presentation and the motivation and also acknowledging the strong experimental results across multiple datasets and tasks over contemporaneous baselines. We proceed here below in replying to the comments in specific.
>
> **W1: The contributions are somewhat overclaimed.**
>
> **A1**. We study active learning for ICL in the practical low-resource scenario, and highlight the importance of diversity-based uncertainty sampling. All AdaICL variants (AdaICL-base, AdaICL, AdaICL) are shown to be effective for ICL, while our optimization formulation (Section 4.2 and 4.3) provides a new framework for combining uncertainty and diversity sampling.
>
> **W2: Technical contributions are not sufficient.**
>
> **A2**. AdaICL performs subset selection based on the hard examples for the LLM, and solves a submodular maximization (MaxCover) problem that captures interactions of different hard examples and their effect during retrieval-based ICL. We believe these are important technical contributions, with connections to previous AL works for kNN classifiers [1].
>
> [1] Wei, Kai, Rishabh Iyer, and Jeff Bilmes. "Submodularity in data subset selection and active learning." ICML, 2015. [(link)](xhttp://proceedings.mlr.press/v37/wei15.pdf)
>
> **W3: The writing also should be polished. For the current form, there are a series of unclear justifications.**
>
> **A3**. We hope that our response addresses the Reviewer's comment and provides more clarifications. We will update the manuscript accordingly.
>
> **Q1: It is possible to meet outliers if the method overemphasizes the selection of diverse data. However, in the current form, it is not clear how to address the issue.**
>
> As we mention in Section 4.1, AdaICL-base uses kmeans clustering that may be affected by outliers. We then propose AdaICL (Section 4.2) that mitigates the effect of outliers by preferring representative examples of dense regions. Finally, AdaICL+ (Section 4.3) aims at avoiding some failing cases of AdaICL, e.g., the case in Figure 9 (Appendix).
>
> **Q2: Does the k-NN retriever equal the similar retriever in the Vote-k paper?**
>
> Exactly, all the methods (Random, Votek, AdaICL variants) use the exact same retriever during inference (e.g., retrieval based on SBERT embeddings). Tables 2 and 8 show that AdaICL is robust with respect to the retriever employed.
> Please also refer to “General Response: Pipeline of the compared methods”.
>
>
> **Q3: This paper claims that "However, these approaches do not consider which examples help the LLM learn new information and may waste resources for annotating examples whose answers are already within the model’s knowledge." I am somewhat confused about this claim. The method Vote-k also uses the feedback of LLMs. Could the paper give more details about this?**
>
> Votek does not perform uncertainty sampling, but uses the LLM’s feedback to partition the examples into $B$ bins (budget equals to $B$) and selects one example from each bin. This might select easy examples for the model, which lead to poor calibration (as it is shown in Section 6.3) or waste annotation resources (Table 1).  In contrast, AdaICL solves a MaxCover problem iteratively, which is adapted to the LLM’s feedback, and does not assume any predefined data partition, e.g., to $B$ bins.

---

> ### Author Response · Authors · 2023-11-15
> **Response to Reviewer LBWT (2/2)**
>
> **Q4: Does the uncertain with respect to one example equal that the LLM can learn it accurately?**
>
> We optimize the MaxCover problem, which assumes that if we annotate an example that the model is uncertain about, it will provide new information to the LLM which can improve its predictions on its neighboring examples (hard examples included in its region).
>
> **Q5: The paper argues that previous work assumes a high-resource setting, where a large set of ICL examples is already annotated. Could the paper provide some detailed examples for better understanding?**
>
> In this work, we focus on the “cold-start” problem, similar to Votek, where we are given an unlabeled set to select examples from. Most of the related works in AL for ICL assume they are given an annotated validation set, which needs to be statistically large and is not practical for real world few-shot applications [2]. The validation set is leveraged for measuring the informativeness of each individual example as well as for hyperparameter tuning. For example, [3] employs reinforcement learning, which requires one set of labeled examples for policy training and another set of labeled examples for reward estimation.
>
> [2] Ethan Perez, Douwe Kiela, and Kyunghyun Cho. “True few-shot learning with language models”. NeurIPS, 2021. [(link)](https://arxiv.org/pdf/2105.11447.pdf) \
> [3] Zhang, Yiming, Shi Feng, and Chenhao Tan. "Active example selection for in-context learning." EMNLP, (2022). [(link)](https://arxiv.org/pdf/2211.04486.pdf)
>
> **Q6: For Section 4.1, could the paper provide more details about how to obtain the probability with respect to the label or demonstrations?**
>
> We follow previous works (MetaICL, Votek), but we would like to make our writing more clear: We feed the model by concatenating the kNN retrieved demonstrations $ [ (x_1, y_1), … (x_k, y_k), x_{k+1} ] $ to compute the negative log-probability for each label $y \in \mathcal{Y}$ for $x_{k+1}$. We keep the prediction with the highest negative log-probability (model’s predicted label with highest confidence) and use its negative log-probability as the uncertainty score. We select the examples with the highest uncertainty scores for the set $U_h$.
>
> **Q7: Compared with previous work such as Vote-k, the method proposed by this work is more complex. It introduces a series of hyper-parameters. How to balance them in practice? Also, is there a time advantage of the proposed method over baselines?**
>
> Great questions, we have proposed a heuristic-based rule (Appendix A.3) for determining these hyper-parameters in practice. Moreover, AdaICL-base does not have many hyper-parameters (apart from the number of clusters $K$ and the uncertainty threshold $\theta$), but outperforms Votek in many tasks (Tables 6 and 7).
>
> **Q8: What is the definition of "egonet" in this paper?**
>
> The egonet is the set of nodes that connect to a central node within N-hops. In our work, the hard region $S_i$ centered around node $i$ includes either hard nodes within 1-hop or within 2-hops.  We will omit the wording “egonet” in the new version to avoid any confusion.
>
> **Q9: For Figure 5, could the paper supplement the comparison between all methods not just the best baseline?**
>
> Yes, thank you for the suggestion. We are working on collecting the experimental results and will update the manuscript accordingly before the rebuttal deadline.

---

### Official Review · Reviewer_6M3N · 2023-10-22

**Soundness:** 3 good
**Presentation:** 3 good
**Contribution:** 3 good
**Rating:** 6
**Confidence:** 3

**Summary:**

This paper presents a method to select samples to present for ICL based on set coverage in an embedding space. The proposed method, AdaICL, uses a greedy approximation for the MaxCover problem to select sets that cover as many hard problems as possible. AdaICL outperforms baselines on a wide variety of tasks and does not appear to be too expensive to run.

I think this paper is closer to a 7 than 6 but unfortunately 7 is not an option on the rating scale.

**Strengths:**

- Good empirical performance on a wide variety of LLMs and datasets.
- AdaICL appears to outperform kMeans based retrievers such as Votek and AdaICL base
- AdaICL appears to be more robust to sample presentation order than the baselines on some tasks

**Weaknesses:**

- Why is the semantic similiarity space determined by a 3rd party embedding model such as SBERT? Shouldn't this be from the LLM itself, such as from an embedding layer?
- How does overall performance depend on $m$ in $G_m$?
- My understanding is that AdaICL queries a LLM many times to get confidence scores *before* feeding the final prompt in to get $y_{test}$. How do the various baselines and AdaICL compare when limited to the same LLM query budget? For example, random selection requires 0 queries, which could be far cheaper than running AdaICL.

**Questions:**

See above.

---

> ### Author Response · Authors · 2023-11-15
> **Response to Reviewer 6M3N (1/1)**
>
> We thank the Reviewer for their appreciation of the strong experimental results across multiple datasets and tasks over contemporaneous baselines as well as the robustness of AdaICL. We proceed here below in replying to the comments in specific.
>
> **W1: Why is the semantic similiarity space determined by a 3rd party embedding model such as SBERT? Shouldn't this be from the LLM itself, such as from an embedding layer?**
>
> **A1**. Thank you for raising this point. We use 3rd party embeddings due to their applicability in practice. First, small-scale LMs, such as SBERT, are much faster at computing embeddings, compared to the LLMs with billions of parameters. Second, for many black-box LLMs, we do not have access to their intermediate layers (or parameters) but only to their outputs (predictions and token logprobabilities). Finally, we have performed experiments with different embedding models (Table 2 and 8), which show that AdaICL is robust to the embedding model selection.
>
> **W2: How does overall performance depend on m in $\mathcal{G}_m$?**
>
> **A2**. We have performed extensive AdaICL ablation studies in Appendix D. Table 11 shows the overall performance with respect to different m values in G_m. In addition, Table 12 compares alternative ways of constructing $\mathcal{G}$.
>
> **W3: My understanding is that AdaICL queries a LLM many times to get confidence scores before feeding the final prompt in to get $y_{test}$ . How do the various baselines and AdaICL compare when limited to the same LLM query budget? For example, random selection requires 0 queries, which could be far cheaper than running AdaICL.**
>
> **A3**. This is an important point. All model-based selection methods (“Hardest”, Votek, AdaICL variants) query the LLM on all candidate examples to get their confidence scores. We have performed a time analysis in Appendix C5 for the example selection process (Random selection has zero cost). With the downsampled version of data, AdaICL outperforms Random without involving many LLM queries.
> During inference, all methods (Random, Votek, AdaICL) have the same cost.
> Please also refer to “General Response: Pipeline of the compared methods”.

---

### Official Review · Reviewer_m5zU · 2023-11-01

**Soundness:** 3 good
**Presentation:** 3 good
**Contribution:** 3 good
**Rating:** 5
**Confidence:** 3

**Summary:**

The authors propose an example selection method for in-context learning via active learning techniques. Given an unlabeled set $\mathcal{U}$ of examples, the authors first choose the set of hard examples $\mathcal{U}_h$ based on model's confidence score $u_i$. Then choose examples up to a given budget $B$ from centroids of each k-mean cluster, which is termed as $\texttt{AdaIcl-base}$. To improve the method, the Maximum Coverage problem is applied. The goal is then to choose $B$ most representative example that cover the most semantic space by building a global graph based on the semantic embedding space ($\texttt{AdaIcl}$). To further improve the method, hard examples are selected from denser regions (instead of outliers) by implementing the re-weighting schema ($\texttt{AdaIcl+}$). The results show effective improvement over nine datasets and seven LLM models.

**Strengths:**

+ The authors provide an intuitive approach which does make sense with additional efficiency.
+ The paper is well structured.
+ Results over many models and datasets showing great performance of variants of $\texttt{AdaICL}$.

**Weaknesses:**

- The construction of the graph is highly dependent on other off-the-shelf encoders, which might not be representative of the target model.
- The method requires multiple prompts to get LLM feedbacks (probability scores) for each example, which is expensive.
- Quite outdated models used (even given the ICLR submission date), so it is hard to verify if the method is applicable to more up-to-date LLMs.

**Questions:**

- How are you performing k-mean clustering for $\texttt{AdaIcl-base}$?

- What embedder are you using for choosing top-k examples?

- Have you compared to any similarity based baselines?

- Have you compared to any retriever-based baselines?

- Question about the practical setting. If we need to annotate the selected examples on demand based on the query, then why not annotate directly the query?

- Why are you choosing top-$N_{\theta}$ examples based on probability scores? Would uncertain examples mean bottom-$N_{\theta}$?

- RQ4. is provided but not addressed in the main paper nor referred to Appendix?

---

> ### Author Response · Authors · 2023-11-15
> **Response to Reviewer m5zU (1/2)**
>
> We thank the Reviewer for their appreciation of the well structured presentation of the paper and for acknowledging the strong experimental results. We proceed here below in replying to the comments in specific.
>
> **W1: The construction of the graph is highly dependent on other off-the-shelf encoders, which might not be representative of the target model.**
>
> **A1**. AdaICL’s graph construction depends on embedding space, which is also true for other methods that use semantic diversity (kmeans, Votek, AdaICL-base). Tables 2 and 8 show that AdaICL is robust to the selection of the embedding model. Moreover, as raised by reviewer 6M3N, we use 3rd party embeddings due to their applicability in practice. First, small-scale LMs, such as SBERT, are much faster at computing embeddings, compared to the LLMs with billions of parameters. Second, for many black-box LLMs, we do not have access to their intermediate layers (or parameters) but only to their outputs (predictions and token logprobabilities).
>
> **W2: The method requires multiple prompts to get LLM feedbacks (probability scores) for each example, which is expensive.**
>
> **A2**. This is an important point. All model-based selection methods (“Hardest”, Votek, AdaICL variants) query the LLM on all candidate examples for annotation to get their confidence scores. We have performed a time analysis in Appendix C5 for the example selection process (Random selection has zero cost). With the downsampled version of data, AdaICL outperforms Random without involving many LLM queries. During inference, all methods (Random, Votek, AdaICL) have the same cost. \
> Please also refer to “General Response: Pipeline of the compared methods”.
>
>
> **W3: Quite outdated models used (even given the ICLR submission date), so it is hard to verify if the method is applicable to more up-to-date LLMs.**
>
> **A3**. Our experiments (Figs. 4 and 6) show that our AdaICL selection strategy generalizes across different LLMs with varying sizes, which --we believe-- make AdaICL’s contributions clear.

---

> ### Author Response · Authors · 2023-11-15
> **Response to Reviewer m5zU (2/2)**
>
> **Q1: How are you performing k-mean clustering for AdaICL-Base?**
>
> During the selection process, we select which examples to annotate (set $\mathcal{L}$) by (i) identifying hard examples for the model (default is 50% most uncertain) , (ii) cluster them into $K$ clusters with $K=B$ ($B$ is the budget), and (iii) selecting the examples closest to each cluster centroid to be added to $\mathcal{L}$. The algorithm is summarized in Appendix A.1.
>
> During inference, set $\mathcal{L}$ is used for retrieval-based ICL. To compare AdaICL-base with random selection, Random creates the set $\mathcal{L}$ drawing $B$ examples at random. Inference is the same for both AdaICL-base and Random ($k$-shot retrieval), but they differ during the selection process.
>
> **Q2: What embedder are you using for choosing top-k examples?**
>
> The default embedder is SBERT, and is the same for all competing methods during retrieval-based ICL. Tables 2 and 8 provide additional ablations on the embedder used, verifying AdaICL’s robustness.
>
> **Q3: Have you compared to any similarity based baselines?**
>
> We are not sure which similarity based baselines the Reviewer refers to. We include diversity-based selection (such as fast-votek). During inference, all methods (Random, Votek, AdaICL) will retrieve the $k$ most similar examples from the annotated set $\mathcal{L}$ as demonstrations for a test query $x_{test}$ .
>
>
> **Q4: Have you compared to any retriever-based baselines?**
>
> Yes, all compared methods use retrieval during ICL inference (including Random, Votek, AdaICL), which has been shown to be the most effective for ICL [1]. We would like to clarify on the pipeline of the compared methods: All methods have a “Selection Phase”, which selects which examples to annotate, and an “Inference Phase”, which is the $k$-shot retrieval-based ICL and is the same for all.
> We summarize the key differences between methods below:
> | Method | Selection |  Inference  |
> | ------------------- | -------------------  | ------------------- |
> | | (how $\mathcal{L}$ is constructed, $B$ examples) |  (using examples from $\mathcal{L}$ for ICL) |
> | Random | random | $k$-shot  ($k \ll B$) retrieval |
> | Hardest | uncertainty | $k$-shot  ($k \ll B$) retrieval |
> | Votek | votek | $k$-shot  ($k \ll B$) retrieval |
> | AdaICL | AdaICL | $k$-shot  ($k \ll B$) retrieval |
>
> Please also refer to “General Response: Pipeline of the compared methods”.
>
> [1] Katerina Margatina, Timo Schick, Nikolaos Aletras, and Jane Dwivedi-Yu. "Active learning principles for in-context learning with large language models", 2023. [(link)](https://arxiv.org/pdf/2305.14264.pdf)
>
> **Q5: Question about the practical setting. If we need to annotate the selected examples on demand based on the query, then why not annotate directly the query?**
>
> As we note above, the selection phase (which examples we choose to annotate) and the inference phase (downstream ICL) are two distinct phases. There is no annotation involved during inference.
>
>
> **Q6: Why are you choosing top-$N_{\theta}$ examples based on probability scores? Would uncertain examples mean bottom-$N_{\theta}$ examples?**
>
> Thank you for raising this. Similar to previous works (MetaICL, Votek), we transform probabilities to negative log-probabilites $u_i$.  Higher $u_i$ means higher uncertainty, that is why we choose top-$N_{\theta}$. We will add this important clarification to the paper.
>
>
> **Q7: RQ4. is provided but not addressed in the main paper nor referred to Appendix?**
>
> Thank you for raising this, there is a typo in the title of Section 6.3, it should be “RQ4: …”. In this section, we investigate how well the model understands the task through the lens of its calibration, i.e., whether higher confidence correlates with higher accuracy. Experimental analysis in Section 6.3 and Appendix C.4, show that AdaICL improves the model’s calibration.

---

> > ### Comment · Reviewer_m5zU · 2023-11-22
> >
> > I appreciate the authors' response.
> >
> > 1. SBERT performance is highly reliant on the selected model. Even for commonly used models like bert-base-uncased and roberta-large, the clustered neighbors exhibit large differences. This problem raises questions about the generalizability of any embedding model. Strangely, the effectiveness of the Random baseline varies for the same dataset, regardless of the embedder employed (as observed in Table 2). Furthermore, Table 8 fails to address variations related to different embedders. Moreover, another limitation arises when relying on SBERT models for obtaining embeddings with a restricted sequence length. This limitation makes the method impractical for processing longer context texts. Thus, the proposed approach seems to be suited only for simpler datasets characterized by shorter contextual information.
> >
> > 2. You have not addressed the problem of expensive multiple querying of the LLMs. Also Table 11 shows that AdaICL (T=2) requires double time compared to Vote k.
> >
> > 3. You have also not addressed the answer directly. Based on the results, I agree with Reviewer LBWT that AdaICL requires hyperparameter tuning. Your observations indicate that different adaptations of AdaICL demonstrate different performance on different datasets. Additionally, different hyperparameters applied to the graph result in varying results across different datasets.

---

> > > ### Author Response · Authors · 2023-11-22
> > > **Response to Reviewer m5zU (Cont.)**
> > >
> > > Thank you again for raising these important points. We would like to address your comments:
> > >
> > > **(1) Embedding models.** The embedding models here are used (A) to determine semantic diversity during active learning (Fast-votek, Votek, Kmeans, AdaICL-base, AdaICL, Patron) and (B) to retrieve the $k$-shot most similar demonstrations during ICL inference.
> > >
> > > (1A) As we mention in Appendix H, most methods rely on embedding methods to determine
> > > semantic diversity (except for Hardest and Random). While AdaICL is shown to be the most robust, we agree that all these methods can still suffer if the semantic space is not accurate for the task. For example, when sequence length is critical or for symbolic tasks like concatenating the last letters of a sequence [1]. In this work, we assume that semantic diversity matters, which is widely assumed in a gamut of NLP tasks. Our assumption is also empirically verified as Hardest and Random methods perform the worst throughout the experiments.
> > >
> > > (1B) Retrieving $k$-shot examples during ICL inference with embedding models, such as SBERT, is currently the state-of-the-art [2], and its importance is also highlighted in theory (see our Section 3, Eq.(2)). Of course, different embedders (SBERT/RoBERTa) affect performance, even when using random selection for annotating examples, as the similarity space during inferenece changes. We add a case where the $k$-shot demonstration for a test instance are determined randomly, which clearly performs the worst, averaged over TREC/SST2/Amazon datasets:
> > > | Selection Method | Inference Retriever |   Inference Retriever |   Inference Retriever  | Inference Retriever  |
> > > | ------------------- | :-------------------: | :-------------------:  | :-------------------: | :-------------------: |
> > > | | Random | SBERT|  RoBERTa | BERT |
> > > |Random |     50.76 | 65.70 |  65.30 | 62.53 |
> > > |Votek | 47.26 |67.87 | 66.01 | 68.95|
> > > |AdaICL| 50.17 | 70.91 | 68.44 | 69.85|
> > >
> > > [1] Wei, Jason, et al. "Chain-of-thought prompting elicits reasoning in large language models." NeuriPS (2022). \
> > > [2] Katerina Margatina, Timo Schick, Nikolaos Aletras, and Jane Dwivedi-Yu. "Active learning principles for in-context learning with large language models", 2023.
> > >
> > > **(2) Cost of querying the LLM.** All model-based active learning methods  (Votek, AdaICL-base, AdaICL, Patron, Hardest) query the LLM to perform uncertainty sampling. This is important as active learning adapts to the LLM used, in contrast to approaches that are independent of the LLM used (Kmeans). Querying the LLM takes place in "Selection Phase" only (see our updated Appendix C), which we believe is a reasonable cost before inference/deployment. The inference procedure is identical for all methods and the selection time is an one time overhead (less than 200secs) - which drastically improves performance (e.g., AdaICL performs 70.95%, while Random performs 59.47% at AGNews). In addition, in the "multi-step" setting (Figures 5, 10), we have AdaICL(T=1), which is shown to outperform all other model-based methods (Votek, AdaICL-base, Patron, Hardest); all these methods have same selection cost.
> > >
> > > **(3) AdaICL hyperparameters.** AdaICL introduces more parameters, compared to AdaICL-base, as it formulates the MaxCover problem, which specifically aims at capturing the effect of different examples during retrieval-based ICL. We believe our main contribution is in the formulation of the MaxCover problem and to facilitate reproducibility, (i) we have the default hyper-parameters throughout the main experiments (Section 5, AdaICL), and (ii) we propose a heuristic rule for determining MaxCover's hyperparameters easily (Appendix B.3). Finally, we note that AdaICL can achieve higher performance with more careful parameter selection (Table 14, AdaICL+ ($\theta = 0.33$)), although we do not report these numbers in Section 6.

---

### Official Review · Reviewer_hvtd · 2023-11-05

**Soundness:** 2 fair
**Presentation:** 3 good
**Contribution:** 1 poor
**Rating:** 3
**Confidence:** 4

**Summary:**

This paper proposes an active learning approach for ICL, which combines diversity-based sampling and uncertainty-based sampling. It introduces three versions of the proposed framework, including ADAICL-BASE, ADAICL, and ADAICL+. The base version performs k-means clustering over the identified hard examples, while ADAICL quantifies whether each example can help the model learn new information, and the plus version further equips a reweighting schema for the MAXCOVER problem to ensure dense regions with hard examples are preferred. Experiments study nine NLP datasets and GSM8K, with 1.3B to 65B LLMs across several model families.

**Strengths:**

1. The paper is presented in a coherent manner and easy to follow.
2. The logical progression connecting the various methodological variants is well-articulated.
3. The improvements over the baseline is good.

**Weaknesses:**

1. Active learning for NLP is a well-studied area. This paper lacks the illustration of why AL for ICL is challenging or the key difference compared to AL for fine-tuning based NLP. Otherwise, why do not directly apply multiple sophisticated query policies proposed in AL to the ICL example selection problem?

2. Although not for ICL, combining diversity and uncertainty for data selection have been studied in previous literature:\
Entropy-Based Active Learning for Object Detection With Progressive Diversity Constraint;\
Cold-start data selection for few-shot language model fine-tuning: A prompt-based uncertainty propagation approach;\
ACTUNE: Uncertainty-Based Active Self-Training for Active Fine-Tuning of Pretrained Language Models.

3. The baseline methods are limited. Although the related work discusses a batch of work for active learning in ICL/NLP, only a few simple baselines are compared in experiments. How does ADAICL compare to other AL methods? Meanwhile, I am also wondering about the performance of zero-shot GPT-4 or GPT-3.5-turbo.

4. The method relies on the estimation of model uncertainty, which is only suited for the LLMs with moderate scales, For those most recent LLMs with hundreds billions parameters, usually we do not have a way to obtain its uncertainty.

**Questions:**

See above.

---

> ### Author Response · Authors · 2023-11-15
> **Response to Reviewer hvtd (1/3)**
>
> We thank the Reviewer for their appreciation of our paper's presentation and the progression of our methodological variants. We proceed here below in replying to the comments in specific.
>
> **W1: Challenges in active learning (AL) for ICL, compared to fine-tuning based NLP**
>
> **A1**:
>  Recent theoretical works relate ICL with nonparametric kernel regression [1,2], which is similar to $k$NN classifiers. Designing AL for non-parametric classifiers has been recently highlighted to be challenging [3], as new information cannot be directly incorporated into the model’s parameters. The seminal work for AL for $k$NN classifiers [4] performs subset selection (set of hard examples) and maximizes a submodular function to capture the interactions of different examples and their effect on the non-parametric classifier. Our AdaICL method follows a similar framework, but employs a MaxCover optimization which can generalize to both classification and generation tasks, as it does not require a finite set of label classes.
>
> Furthermore, although active learning for NLP is well-studied, most of the recent approaches fine-tune the model during different AL rounds. This allows the model to incorporate information from the newly labeled examples into its parameters, which can gradually improve its predictions. However, LLMs with billions of parameters are used for ICL. In this case, computing gradient updates is costly and requires additional fine-tuning  for every new task. Finally, ICL with wisely-selected labeled samples is shown to be a better few-shot practice than supervised finetuning ([5], Section 4.1).
>
> [1] Chi Han et al., "In-context learning of large language models explained as kernel regression", 2023. [(link)](https://arxiv.org/pdf/2305.12766.pdf) \
> [2] Yu Bai et al., "Transformers as statisticians: Provable in-context learning with in-context algorithm selection", 2023. [(link)](https://arxiv.org/pdf/2306.04637.pdf) \
> [3]: Rittler, Nicholas, and Kamalika Chaudhuri. "A two-stage active learning algorithm for k-nearest neighbors." ICML, 2023. [(link)](https://arxiv.org/pdf/2211.10773.pdf) \
> [4]: Wei, Kai, Rishabh Iyer, and Jeff Bilmes. "Submodularity in data subset selection and active learning." ICML, 2015. [(link)](http://proceedings.mlr.press/v37/wei15.pdf) \
> [5] Su, Hongjin, et al. "Selective annotation makes language models better few-shot learners." ICLR, 2023. [(link)](https://arxiv.org/pdf/2209.01975.pdf)

---

> ### Author Response · Authors · 2023-11-15
> **Response to Reviewer hvtd (2/3)**
>
> **W2: Related work combining diversity and uncertainty sampling**
>
> **A2**. We thank the reviewer for bringing forward these important related works. We will make sure to include them in our Related Work section of the updated manuscript.
>
> We note that [2] is specifically designed for object detection and cannot be trivially adapted for NLP, while [3] aims at selecting the best pseudo-labels (where budget B=0), which is not optimal for retrieval-based ICL as our experiments show (Table 2 “Pseudo-labeling”).
> We highlight the key differences between our work (AdaICL) and Patron [1], which also targets for selective annotation at the cold-start problem:
>
> *Optimization*. Patron computes an uncertainty score (based on model’s feedback)  and a diversity score (based on kmeans clustering) for each example (Eq.(8)/(10) in [1]). Then, it selects the examples with the highest aggregate score from each cluster (similar to our AdaICL-base).
> In our AdaICL method, we solve MaxCover (a submodular function maximization) over the data subset with hard examples.  The MaxCovar avoids selecting examples that will have similar effect during retrieval-based ICL, which leads to a dynamic (NP-hard) optimization. This optimization does not depend on pre-defined clusters, but focuses on examples that can convey new information to the model during inference, even if they would belong to the same cluster.
>
> *Multi-step adaptation*. At each step (AL round), AdaICL estimates the model’s uncertainty and solves MaxCover over the new subset with hard examples. As the sets of hard examples may be completely different during successive steps, AdaICL can capture broad interactions of the examples and their effect during ICL. Patron combines uncertainty and diversity in a linear manner, which may lead to preferring examples with specific properties (e.g., ones that are close to the cluster centers).
>
> To empirically verify our hypothesis, we adapt Patron to our tasks, and compare AdaICL with Patron in the multi-step setting (Figure 5):
> |  Topic Classif.(AGNews, TREC) | B=5  |  B=15 |  B=25 |
> | -------- | ------- |  ------- |  -------- |
> | Patron  | 39.11 | 39.35 | 49.64 |
> | AdaICL | 38.89 | 52.40 | 62.57 |
>
> |  Sent. Analysis (SST2, Amazon) | B=5  |  B=15 |  B=25 |
> | -------- | ------- |  ------- |  -------- |
> | Patron  | 61.45 | 74.75 | 79.79 |
> | AdaICL | 61.38 | 79.35 | 80.51 |
>
> We believe these to be good results and that they positively address the Reviewer’s request. Moreover, our results show that Patron is not suitable for tasks where semantic diversity is less important, i.e., RTE, MRPC, and MNLI:
>
> |  Nat. Lang. Inf. | RTE  |  MRPC |  MNLI |
> | -------- | ------- |  ------- |  -------- |
> | Random  | 48.30 | 64.48 | 40.99 |
> | Patron | 48.16 | 65.04 | 38.89 |
> | AdaICL | 53.12 | 67.05 | 42.96 |
>
> [1] Yu, Yue, et al. "Cold-start data selection for few-shot language model fine-tuning: A prompt-based uncertainty propagation approach." ACL, 2023. [(link)](https://arxiv.org/pdf/2209.06995.pdf) \
> [2] Wu, Jiaxi, Jiaxin Chen, and Di Huang. "Entropy-based active learning for object detection with progressive diversity constraint." CVPR. 2022.  [(link)](https://arxiv.org/pdf/2204.07965.pdf) \
> [3] Yu, Yue, et al. "AcTune: Uncertainty-based active self-training for active fine-tuning of pretrained language models." NAACL. 2022. [(link)](https://aclanthology.org/2022.naacl-main.102.pdf)

---

> ### Author Response · Authors · 2023-11-15
> **Response to Reviewer hvtd (3/3)**
>
> **W3: Baseline methods for active learning in ICL**
>
> **A3**. In this work, we focus on the “cold-start” problem, similar to Votek, where we are given an unlabeled set to select examples from. Most of the related works in AL for ICL assume they are given an annotated validation set, which needs to be statistically large and is not practical for real world few-shot applications [1]. The validation set is leveraged for measuring the informativeness of each individual example as well as for hyperparameter tuning. For example, [2] employs reinforcement learning, which requires one set of labeled examples for policy training and another set of labeled examples for reward estimation. Moreover, as suggested, we compare AdaICL with Patron, showcasing the benefits that AdaICL offers for ICL.
>
> GPT-4 or GPT-3.5-turbo are not open-sourced and we currently do not have access to. However, we agree that comparing (i) smaller LLMs combined with AL against (ii) larger LLMs with zero-shot is an important direction.
>
> [1] Ethan Perez, Douwe Kiela, and Kyunghyun Cho. “True few-shot learning with language models”. NeurIPS, 2021. [(link)](https://arxiv.org/pdf/2105.11447.pdf) \
> [2] Zhang, Yiming, Shi Feng, and Chenhao Tan. "Active example selection for in-context learning." EMNLP, (2022). [(link)](https://arxiv.org/pdf/2211.04486.pdf)
>
> **W4: Obtaining model uncertainty for LLMs**
>
> **A4**. We can obtain uncertainty for large  black-box LLMs through their APIs. For example, ChatGPT provides the `logprobs` outputs, while other LLMs have other [ways](https://huggingface.co/bigscience/bloom/discussions/89). AdaICL can be applied to any LLM from which we can access its output logits/probabilities (we discuss this in Appendix F).

---

### Author Response · Authors · 2023-11-15
**General Response**

We thank the reviewers for their valuable feedback. We would like to address the main points raised, before addressing comments from each reviewer independently. During the rebuttal period, we will update the manuscript to include the clarifications and new experimental results.

**1) Pipeline of the compared methods** (Reviewers m5zU, LBWT) **and their complexity** (Reviewers 6M3N).

We would like to clarify on the pipeline of all compared methods. All methods have a “Selection Phase”, which selects which examples to annotate, and an “Inference Phase”, which is the k-shot retrieval-based ICL and is the same for all.  We use $k$-shot retrieval-based inference as it is shown to be more effective for ICL [1].  We provide the following comparison Table, where compared methods differ during the “Selection Phase”:
| Method | Selection |  Selection Complexity |  Inference  |
| ------------------- | :-------------------: | :-------------------:  | :-------------------: |
| | (how $\mathcal{L}$ is constructed, $B$ examples) |  | same for all (using examples from $\mathcal{L}$ for ICL) |
| Random | Random | Zero-cost | $k$-shot  ($k \ll B$) retrieval |
|Kmeans | Clustering | Independent of the LLM |  $k$-shot  ($k \ll B$) retrieval |
| Hardest | Uncertainty | Depends on LLM's complexity |  $k$-shot  ($k \ll B$) retrieval |
| Votek | Votek | Depends on LLM's complexity | $k$-shot  ($k \ll B$) retrieval |
| AdaICL | AdaICL | Depends on LLM's complexity |  $k$-shot  ($k \ll B$) retrieval |


As it is shown, all methods use the same retriever during inference and have the same computation cost. During selection, model-based methods (Votek, AdaICL) have a higher cost, but this cost is only needed before inference/deployment.  We have performed a time analysis in Appendix C5 for the example selection process: With the downsampled version of data, AdaICL outperforms Random without involving many LLM queries. During inference, all methods (Random, Votek, AdaICL) have the same cost.

[1] Katerina Margatina, Timo Schick, Nikolaos Aletras, and Jane Dwivedi-Yu. "Active learning principles for in-context learning with large language models", 2023. [(link)](https://arxiv.org/pdf/2305.14264.pdf)


**2) Related works** (Reviewers hvtd, LBWT).

In this work, we focus on the “cold-start” problem, similar to Votek, where we are given an unlabeled set to select examples from. Most of the related works in AL for ICL assume they are given an annotated validation set, which needs to be statistically large and is not practical for real world few-shot applications [1]. The validation set is leveraged for measuring the informativeness of each individual example as well as for hyperparameter tuning. For example, [2] employs reinforcement learning, which requires one set of labeled examples for policy training and another set of labeled examples for reward estimation. Moreover, under our discussion with Reviewer hvtd, we compare AdaICL with an important related work (Patron [3]), showcasing the benefits that AdaICL offers for ICL.

[1] Ethan Perez, Douwe Kiela, and Kyunghyun Cho. “True few-shot learning with language models”. NeurIPS, 2021. [(link)](https://arxiv.org/pdf/2105.11447.pdf) \
[2] Zhang, Yiming, Shi Feng, and Chenhao Tan. "Active example selection for in-context learning." EMNLP, (2022). [(link)](https://arxiv.org/pdf/2211.04486.pdf) \
[3] Yu, Yue, et al. "Cold-start data selection for few-shot language model fine-tuning: A prompt-based uncertainty propagation approach." ACL, 2023. [(link)](https://arxiv.org/pdf/2209.06995.pdf)


**3) Embedding model effect** (Reviewers 6M3N, m5zU).

We use 3rd party embedding models, such as SBERT, instead of the LLM itself due to their applicability in practice. First, small-scale embedding models, such as SBERT, are much faster during computations and inference, compared to the LLMs with billions of parameters. Second, for many black-box LLMs, we do not have access to their intermediate layers (or parameters) but only to their outputs (predictions and token log probabilities). Finally, we have performed experiments with different embedding models (Table 2 and 8), which show that AdaICL is robust to the embedding model selection.

---

### Comment · Area_Chair_oft1 · 2023-11-17
**Author-Reviewer Discussion Phase**

Thank you, reviewers, for your work in evaluating this submission. The reviewer-author discussion phase takes place from Nov 10-22.

If you have any remaining questions or comments regarding the rebuttal or the responses, please express them now. At the very least, please acknowledge that you have read the authors' response to your review.

Thank you, everyone, for contributing to a fruitful, constructive, and respectful review process.

AC

---

### Author Response · Authors · 2023-11-20
**New manuscript revision**

We would like to signal the upload of a new manuscript revision. This includes the changes anticipated in the previous general comment and throughout the responses to each single reviewer.

The revision includes the following main additions (reported in the paper appendix):

1. **Related Works, Section 2**: We added more elaborations on existing related works on ICL and other active learning methods for finetuned NLP (Reviewers hvtd, LBWT). Our previously extended related works are in Appendix A.
2. **Motivation on Sections 3 & 4**: We added more details about the challenges on active learning for ICL, and how AdaICL aims at combining uncertainty with diversity sampling in an effective way for the ICL classifier (Reviewer hvtd). We also elaborate on why we use third-party embedding models (Reviewers 6M3N, m5zU).
3. **Appendix C**: We clarify on the pipeline of the compared methods (Reviewers m5zU, LBWT, 6M3N).
4. **Appendix E.2**: We added full results of Figure 5 (Reviewer LBWT) and comparison with Patron, a sampling method combining uncertainty and diversity sampling (Reviewer hvtd).

When necessary, we added small changes to improve readability and elaborate on certain aspects raised by the reviewers. Changes in the revision are visually highlighted in blue.

---

### Author Response · Authors · 2023-11-22
**A gentle reminder to All Reviewers**

Dear Reviewers,

This is a just a gentle reminder that the discussion period is a day away. We would appreciate it if you could take a look at the general response, our responses to individual reviews and updates to the manuscript. Please let us know if you have any additional questions or concerns, as we would be happy to engage in discussion and address them. Thank you once again for your efforts!

---

### Meta-Review · Area_Chair_oft1 · 2023-12-05

**Metareview:**

This paper studies an interesting problem of in-context learning, which actively selects unlabeled data for annotating and thereby benefits in-context learning. However, there still are several concerns. The comparison methods are limited, and some active learning methods for NLP and ICL, as well as some zero-shot methods of SOTA LLMs, are lacking. Additionally, the essential part of this method, e.g., uncertainty, is hard to estimate for recent LLMs with huge amounts of parameters and close-source LLMs. Therefore, I recommend rejecting this paper, but I encourage the authors to use feedback from reviewers to improve the paper.

**Justification For Why Not Higher Score:**

The reason for not assigning a higher score is due to the following shortcomings identified in this paper: 1. The comparison methods of this paper are weak, which makes the results of this paper not convincing. 2. The practical significance of the setting in this paper needs to be further discussed.

**Justification For Why Not Lower Score:**

N/A

---

### Decision · Program_Chairs · 2024-01-16

Reject